# High performance artificial visual perception and recognition with a plasmon-enhanced 2D material neural network

Tian Zhang [1], Xin Guo[1,2], Pan Wang [1,2], Xinyi Fan[1], Zichen Wang[1], Yan Tong[1], Decheng Wang[1], Limin Tong[1,2] & Linjun Li [1,2] ✉

The development of neuromorphic visual systems has recently gained momentum due to their potential in areas such as autonomous vehicles and robotics. However, current machine visual systems based on silicon technology usually contain photosensor arrays, format conversion, memory and processing modules. As a result, the redundant data shuttling between each unit, resulting in large latency and high-power consumption, seriously limits the performance of neuromorphic vision chips. Here, we demonstrate an artificial neural network (ANN) architecture based on an integrated 2D $MoS_2$/ Ag nanograting phototransistor array, which can simultaneously sense, pre-process and recognize optical images without latency. The pre-processing function of the device under photoelectric synergy ensures considerable improvement of efficiency and accuracy of subsequent image recognition. The comprehensive performance of the proof-of-concept device demonstrates great potential for machine vision applications in terms of large dynamic range (180 dB), high speed (500 ns) and low energy consumption per spike ($2.4 \times 10^{-17}$ J).

The human visual system is mainly composed of the eyes and the visual cortex of the brain[1,2]. The retina of the eye is normally used to capture external optical information and perform first-stage image pre-processing[3–5]. The regulated visual signals are transmitted to the neural network of the visual center for final processing and recognition[6,7]. Accordingly, a variety of bio-inspired artificial visual perception and recognition modules (AVPRM) for emulating certain functions of the human eye and neural network image processing have emerged that are used to perform typical image processing function-alities, which include image-contrast enhancement[1,2,8,9], noise suppression[10,11], visual adaptation[5,12], detection and recognition[13–19], and auto-encoding[20]. In addition, the first prototype of artificial optical graded neuron was proposed and realized for processing spatio-temporal information with more than 99% accuracy[21]. However, for current AVPRM, a hardware solution with both the pre-processing function of the human retina and the image recognition capability of

the visual cortex has not been reported, especially in on-site critical applications[18,20]. There is a high demand to develop multifunctional electronic devices to meet the challenges of next generation machine vision. Additionally, developing low-power and high-efficiency AVPRM has become a major research focus, where the most critical issue to be addressed is the efficient conversion of optical images into electrical digital signals.

Plasmonic energy conversion has been considered as a promising alternative to drive a wide range of physical and chemical processes[22–25]. This emerging method is based on the generation of hot electrons with energy distribution deviating substantially from equili-brium Fermi-Dirac distribution in plasmonic nanostructures after light absorption through non-radiative electromagnetic decay of surface plasmons[26–30]. While the 2D semiconductor itself has excellent optoe-lectronic properties[31–33] such as ultrafast response[34,35], external tunability[20,36] and large photothermoelectric effect[37], plasmonics can

[1]State Key Laboratory of Extreme Photonics and Instrumentation, College of Optical Science and Engineering, Zhejiang University, Hangzhou 310027, China. [2]Intelligent Optics and Photonics Research Center, Jiaxing Institute Zhejiang University, Jiaxing, China. ✉ e-mail: lilinjun@zju.edu.cn

further enable strong light-matter interactions in 2D materials[38,39]. 2D materials technology has by now achieved a sufficiently high level of maturity for integration with conventional complex electronic systems[40–42]. Herein, we present a plasmonic phototransistor array (PPTA) constructed of nanogratings and 2D heterostructures, which constitutes an ANN that integrates simultaneous sensing, pre-processing and image recognition functions. The plasmonic photo-transistor (PPT) takes advantage of the strong coupling of photonic and electronic resonances in an elaborately designed device, in which hot electrons are injected efficiently into the floating gate and produce a large photoelectric effect, to simulate the response of the human retina to optical color information. Moreover, the electrical dynamic modulation of the gate electrode can effectively enlarge the dynamic range of the device for image pre-processing functions (image contrast enhancement). Further real-time image recognition is realized by training the network through varying the drain-source voltage to set the photoresponsivity value of each pixel individually. As a result, the AVPRM integrated with image pre-processing and ANN can effectively improve the image quality, and increase the efficiency and the accuracy of image recognition.

## Results

### The structure and mechanism of PPTA

Figure 1a illustrates the schematic structure of a 2D PPT, which consists of a 2D $MoS_2$/Ag nanograting integrated structure on the left and a 2D $MoS_2$/h-BN/$WSe_2$ heterostructure on the right. The left part of the device mimics the sensing and pre-processing functions of the human retina for color information (Supplementary Fig. 1a) using light-excited waveguide-plasmon polaritons (WPPs)[43] and electrical modulation of the gate electrode, respectively (see Fig. 2 for more details of the mechanism). The photocurrent signal processed in the first stage can be passed to the floating gate on the right side of the device to induce the channel current, which is similar to that visual information can be transmitted through the optical nerve to each neuron in the visual center via synaptic interconnection (Supplementary Fig. 1a, b). The photoresponsivity (synaptic weight) of the device is modulated by changing the drain-source voltage to emulate the regulation of neurotransmitter release between biological synapses (Supplementary Fig. 1c). To avoid unnecessary direct photocurrents in the channel, the right side is covered by the $Al_2O_3$/Au layer. Interconnecting each 2D PPT (subpixel) in the form of an ANN constitutes an AVPRM with image sensing, pre-processing and recognition functions (Fig. 1b). It contains $N$ pixels, which form the imaging array, and each pixel is divided into $M$ subpixels. The circuit connections of $M$ subpixels and $N$ pixels are presented in Fig. 1c, d, respectively. Each subpixel delivers a photocurrent of $I_{mn} = R_{mn}P_n$ under illumination, where $R_{mn}$ is the regularized photoresponsivity of the subpixel and $P_n$ denotes the optical power at the $n$th pixel. $n = 1,2,…,N$ and $m = 1,2,…,M$ denote the pixel and subpixel indices, respectively. Figure 1e depicts the entire operation process of AVPRM in the form of a flowchart. The input optical image is first sensed by the hybrid plasmonic structure in PPT, and the perceived electrical signal is modulated by the side gate electrode in PPT to achieve the pre-processing of the signal. Then, the preprocessed signals are transported to ANN base on a single-layer perceptron, and the network is trained off-line using computer simulation. Subsequently, the predetermined photoresponsivity matrix, that is, photoresonsivities scaled from dimensionless weights, is transferred to the PPTA to complete the image recognition.

The schematic of a classifier is provided in Supplementary Fig. 1d. The array is operated as a single-layer perceptron using pre-processed visual information as the input layer. Here, we chose the softmax function $\phi_m(I) = e^{I_m\xi} / \sum_{k=1}^{M} e^{I_k\xi}$ as the nonlinear activation function to generate the neuron output off-chip, where $\xi = 10^{11}A^{-1}$ is a scaling factor. In one type of ANN representing a supervised learning algorithm, in order to facilitate the classification of images **P** into different categories **y**, we chose a binary code encoding, where each of the three letters corresponds to an output code. Following the elaborated design concept of the 2D PPTA, we fabricated the actual device as shown in Fig. 1f. The sample fabrication process is provided in Supplementary Figs. 2 and 3 (for details, see Methods). This device consists of 27 subpixels ($N \times M = 27$), of which every 9 subpixels were arranged to form a $3 \times 3$ imaging array ($N = 9$) with a subpixel size of about $17 \times 5 \ \mu m^2$. A schematic of the entire circuit connections of the array is presented in Supplementary Fig. 4. Summing all photocurrents generated by 9 PPTs with the same subpixel index $m$ according to Kirchhoff's law, the output $I_m$ is expressed as

$$I_m = \sum_{n=1}^{N} I_{mn} = \sum_{n=1}^{N} R_{mn}P_n \tag{1}$$

Figure 1g shows the high-resolution scanning transmission electron microscope and energy dispersive X-ray spectroscopy element mapping characterizations of a single subpixel in the black box in Fig. 1f, indicating a clean heterostructure interface. The additional analysis on the $MoS_2$, h-BN and $WSe_2$ flakes is described in Supplementary Fig. 5.

In order to understand the mechanism of 2D PPT, we present a scenario for elaboration below. As shown in Fig. 2a, following light absorption and localized surface plasmon resonance (LSPR) excitation in the Ag nanograting, the electromagnetic resonance can be damped radiatively by re-emission of photons, or non-radiatively through transferring the energy to hot electrons via Landau damping[22,26]. In the subsequent hot electron injection[29] (Fig. 2b), hot electrons with momentum within the escape cone[28] can be rapidly emitted into $MoS_2$ through ohmic contacts during the relaxation time[27,39]. At the same time, 2D $MoS_2$ itself also produces a fraction of energetic hot electrons after absorbing light energy, although the effect of this fraction is minimal. The quantitative comparison of the photocurrents of the devices with and without nanograting shows that the plasmon enhancement effect plays a crucial role in the generation and transport of hot electrons (see Supplementary Fig. 6). Figure 2c shows the simulated normalized transmittance mapping of the grating period from 250 to 450 nm in the visible region (for details, see Methods), where Rabi splitting can be clearly observed as a distinguishing characteristic of the strong coupling. It is worth mentioning that the upper, middle and lower three hybrid branches are caused by the coupling of the symmetric and antisymmetric modes in the waveguide with the LSPR mode, respectively, and the bottom two branches are caused by the presence of the mode in the quartz substrate, which is independent of the strong coupling modes. We choose the three eigenenergies corresponding to the red (632 nm), green (535 nm) and blue light (469 nm) when the grating period is 320 nm as the eigenvalues of the three-coupled oscillator model to analyze the strong coupling of this structure. The obtained Rabi splitting ($\Omega \approx 680$ meV) is satisfied with the strong coupling criterion between these three oscillators, that is, $\Omega > \mathbf{W} \cdot \sum_{i=Pl,Sym,Asym} \mathbf{P}^i \gamma_i$, where $\mathbf{W} = (W_{Upper}, W_{Middle}, W_{Lower})$ are the weight of each hybrid branch, $\mathbf{P}^i = (P^i_{Upper}, P^i_{Middle}, P^i_{Lower})$ represents the proportion of uncoupled states in each branch, and $\gamma_i$ represents the linewidth of each uncoupled mode (for details, see Methods). The electric field distribution corresponding to the eigen-energy of different branches at the period 320 nm is provided in Fig. 2d. It can be clearly found that the coupling between LSPR mode and waveguide mode leads to energy exchange. The above mechanism suggests that the 2D PPT can respond to optical color information. Thus, by exploiting the hybrid LSPR and waveguide modes, we realize highly efficient photoelectric conversion, while the limitation on the narrow responding wavelength of LSPR could be surmounted by adjusting the dimension of the Ag nanograting structure.

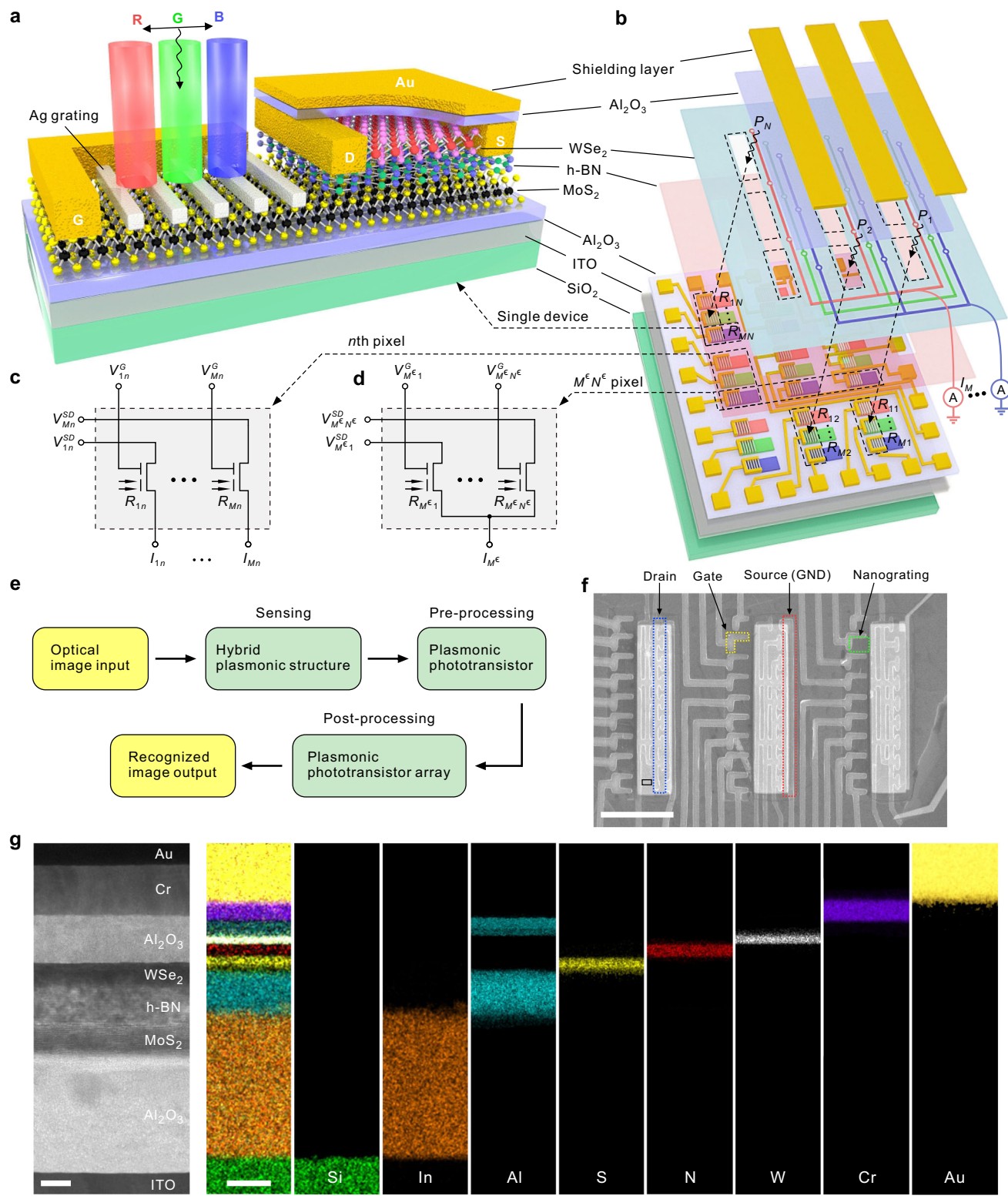

On the other hand, the hot electrons that can not be emitted from the decay of plasmons can generate enormous heat on the picosecond scale, which leads to a balance between thermoelectric potential $E_T$ (Supplementary Fig. 7) and the accumulated electropotential $E_A$ as shown in Fig. 2e, f[37]. The band diagram shown in the lower part of Fig. 2e illustrates this process. The band diagram was divided into two parts, which correspond to the Ag/MoS$_2$ architecture on the left side of the PPT device and the MoS$_2$/hBN/WSe$_2$ architecture on the right side. As shown in the lower part of Fig. 2e, the hot electrons

generated by the decay of plasmons are injected into the conduction band of MoS$_2$, and then the hot electrons are transported to the right side by tilting the energy band under the action of thermoelectric potential. Subsequently, the electrons transported to the right side of MoS$_2$ induced holes in the valence band of WSe$_2$, which were then used for current measurement. Obviously, the higher the optical power, the larger the measured channel current. After the light is turned off (Fig. 2f), the electrons are restored to the initial state by the accumulated potential $E_A$. With such mechanism, the device can respond to

**Fig. 1 | Artificial visual perception and recognition module (AVPRM)-inspired 2D artificial neural network (ANN) plasmonic phototransistor array (PPTA).** **a** Schematic of a 2D PPT. The white G, D and S represent the gate electrode, source electrode, and drain electrode, respectively. **b** Disassembled diagram of the 2D ANN PPTA. The current induced by subpixels of the same color in WSe$_2$ channel layer is connected in parallel by wires of the same color to generate an output current $I_M$. $R_{11}...R_{MN}$ represent the photoresponsivity of each subpixel. $P_1...P_N$ represent the optical power incident on each pixel. Circuit diagram of the $n$th pixel (**c**) and $M^∈N^∈$ subpixels (**d**) in the array, where $M^∈$ is a subset of $M$, representing a certain number of subpixels among $M$ subpixels with the same $M$ indice, and $N^∈$ is a subset of $N$, representing a certain number of pixels among $N$ pixels. $V_{1n}^G...V_{Mn}^G$

represent the gate voltage applied to each subpixel contained i*n* the $n$th pixel. $V_{1n}^{SD}...V_{Mn}^{SD}$ represent the drain-source voltage applied to each subpixel contained in the $n$th pixel. **e** Illustration of an AVPRM based on the 2D PPT for image pre-processing and an ANN for image recognition. **f** Scanning electron microscopy (SEM) image of the PPTA. Scale bar, 20 μm. GND, ground electrode. The blue dashed box shows the drain electrode, the yellow dashed box shows the gate electrode, the red dashed box shows the source electrode, and the green dashed box shows the nanograting. **g** High-resolution scanning transmission electron microscope image captured from the black box in (**f**) and energy dispersive X-ray spectroscopy mapping. Scales bar are 8 (left) and 40 nm (right), respectively.

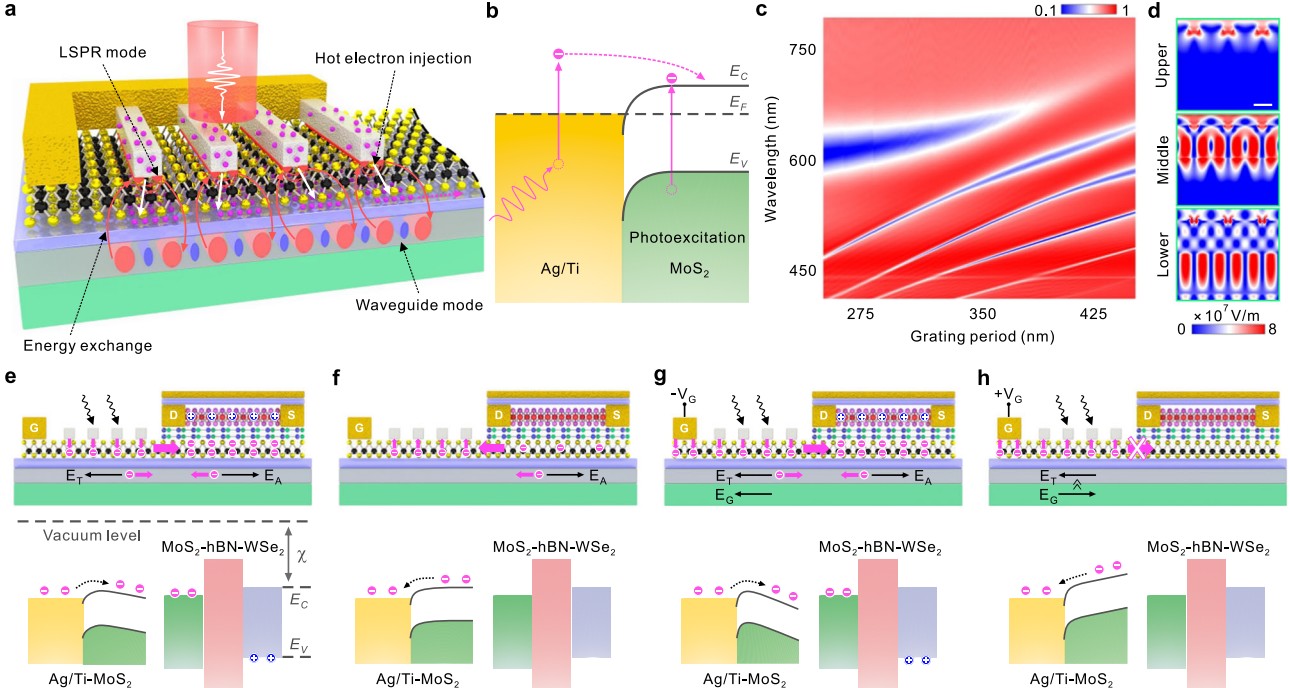

**Fig. 2 | Schematic illustrations of the mechanisms of 2D PPTA. a** A diagrammatic model proposed to describe the whole physical process of the strong coupling between the localized surface plasmon resonance (LSPR) mode and the waveguide mode and its relaxation. The red arrow represents the energy exchange caused by the coupling between LSPR mode and waveguide mode. The white arrow represents hot electron injection from the Ag nanograting to the 2D MoS$_2$. **b** Simplified band diagram illustrating the hot electron injection process taking place at the Ag-MoS$_2$ interface. In addition to receiving hot electrons emitted from the Ag nanograting, MoS$_2$ itself can also generate a small amount of electrons after receiving light. $E_C$, $E_F$, and $E_V$ represent the conduction band, Fermi level and valence band of 2D MoS$_2$, respectively. **c** The simulated transmittance spectrum of the Ag nanograting-ITO waveguide integrated structure dependent on the grating period,

showing the classic Rabi splitting. **d** The calculated electric field distribution at the 320 nm grating period corresponding to each branch at red (R), green (G) and blue (B) wavelengths in the strong coupling regime. Scale bar, 180 nm. Charge-flow illustrations and schematic band diagrams at different operation modes: light on (**e**), light off (**f**), light on and apply $-V_G$ (**g**), light on and apply $+V_G$ (**h**). $V_G$ represents the gate voltage applied to the gate electrode. The blue circles denote the holes, the magenta circles denote the electrons, and the magenta arrows indicate the flow direction of the electrons. $E_T$ represents the thermoelectric potential, $E_A$ represents the accumulated potential, and $E_G$ represents the gate potential. The black arrows represent the direction of each potential. The black dotted arrow represents the direction of the electron transition.

different luminance (gray scale of image). When the light is turned on and the negative side gate voltage $-V_G$ is applied, the electrons will be more easily transferred from the left side of MoS$_2$ to the right side, as there is an additional gate potential $E_G$ (Fig. 2g). From the perspective of energy band (lower part of Fig. 2g), it can be explained that the energy band of MoS$_2$ is more inclined under the combined effect of thermoelectric potential ($E_T$) and gate potential ($E_G$), making it easier for electrons to be transported to the right side. Accordingly, the larger channel current will be induced by the MoS$_2$ gate. Conversely, by applying a positive gate voltage $+V_G$ while the light is turned on, the electrons will be dragged to the left side because of the additional gate potential $E_G$ (Fig. 2h). From the perspective of energy band (lower part of Fig. 2h), in this case, the energy band of MoS$_2$ are tilted in the opposite direction due to the effect of the gate potential ($E_G$), which

makes it difficult for the electrons to overcome this potential to reach the right side. The holes left on the right side of the floating gate lead to electron doping to the channel, which gives low conductance since WSe$_2$ is a p-type semiconductor. The mechanism of the device described in Fig. 2g, h can be used to eliminate the redundant information. Finally, the regulation of the photoresponsivity of a single device can be realized by changing the drain-source voltage, which can be used to train the weights in the ANN formed by interconnected devices.

## Image recognition based on device characterization
Having described the design concept of AVPRM, we next present its feasibility from an experimental perspective. The optical experimental setup is shown in Supplementary Fig. 8a, b and the electrical

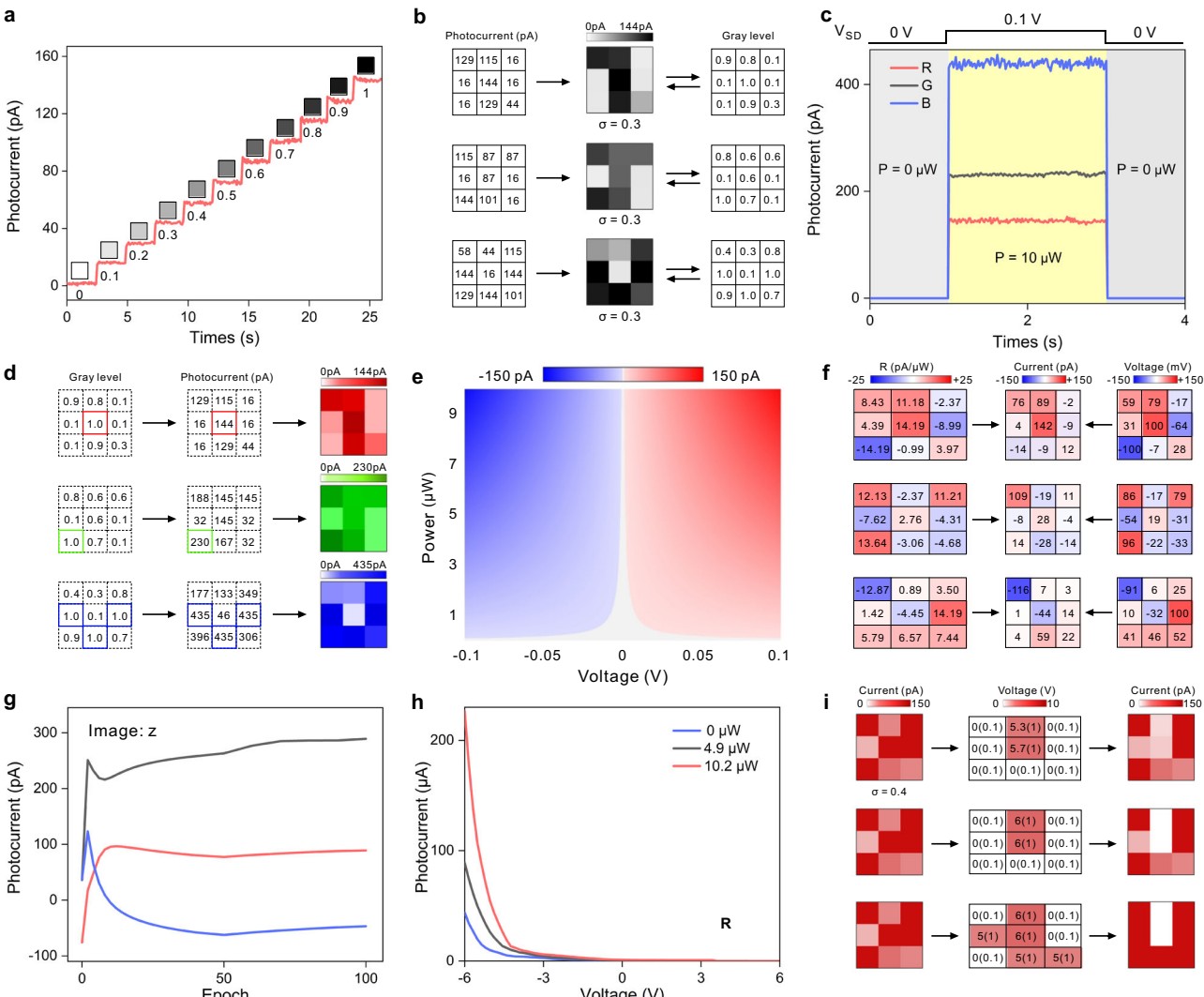

**Fig. 3 | Functional implementation of 2D PPTA. a** Multi-state photocurrents corresponding to different levels of optical power (gray levels), where the laser wavelength is 635 nm and the drain-source voltage is 0.1 V. **b** Extraction of the gray level of each pixel in the image. The images were added with Gaussian noise (with standard deviation of $\sigma = 0.3$). **c** Photocurrent of different colors of light (R: 635 nm, G: 532 nm, B: 473 nm) under the same measurement conditions, where the power is 10 μW and the drain-source voltage is 0.1 V. **d** Recognition of different color images. The numbers in different colored boxes represent the maximum gray level extracted from (**b**) and its corresponding current. **e** The voltage ($V_{DS}$) tunable photocurrent corresponding to each gray level. **f** The photoresponsivity of the

array and its modulation process after 100 off-line training epochs of the letter 'z' in (**d**). **g** The total output current of the projected letter after each training epoch, where the letter 'z' in (**d**) is projected onto the chip. The maximum current corresponds to the label of the projected letter. **h** The transfer characteristic curves of the devices with red light measured under different $P$ values at $V_{DS} = 1$ V, respectively. **i** The pre-processing process of images. The left column represents the image with Gaussian noise ($\sigma = 0.4$) added before pre-processing, the middle column represents the modulation voltage required for pre-processing, and the right column represents the image after pre-processing.

experimental setup is shown in Supplementary Fig. 9a (for details, see Methods). Here we choose the red light of $\lambda = 635$ nm, and its power (0–10 μW) is divided into 11 orders. Figure 3a presents the multi-state photocurrents corresponding to different levels of optical power. These photocurrents are graphically visualized as 11 gray levels in the 0–1 interval. Thus, the gray level of each pixel in the image can be extracted and presented through photocurrent measurement, as shown in Fig. 3b. By measuring the photocurrent corresponding to three wavelengths of light at the same power $P = 10$ μW, we can distinguish red (635 nm), green (532 nm) and blue colors (473 nm) when $V_{DS} = 0.1$ V (Fig. 3c). As shown in Fig. 3d, when the photocurrent of the pixel with the largest gray level in the image is measured, the color of the image can be distinguished by different current values. This is caused by the different absorption rates of the device for the corresponding three wavelengths of light in the strong coupling mechanism

(see Supplementary Fig. 10a). Also, the measurement of transmission spectra of 27 PPTs indicates that the device has good uniformity (see Supplementary Fig. 11). Next, we performed photocurrent-voltage ($I_{PH}$-$V_{DS}$) characteristic measurements under different optical powers (see Supplementary Fig. 10b). It shows a linear dependence of the photocurrent on the voltage over a wide voltage range, which indicates that the device is dominated by ohmic contacts. Then, we extracted photocurrent as a function of optical power under different $V_{DS}$ values (inset in Supplementary Fig. 10c). A nearly symmetrical and adjustable (trainable) linear photoresponsivity between −15 and +15 pA/μW can be obtained by varying the $V_{DS}$ (Supplementary Fig. 10c). Considering the subsequent ANN training, we plotted the voltage tunable photocurrents corresponding to each gray level, as shown in Fig. 3e. Similar measurements of the optoelectronic characterization of green and blue light and the uniformity of each device are presented in

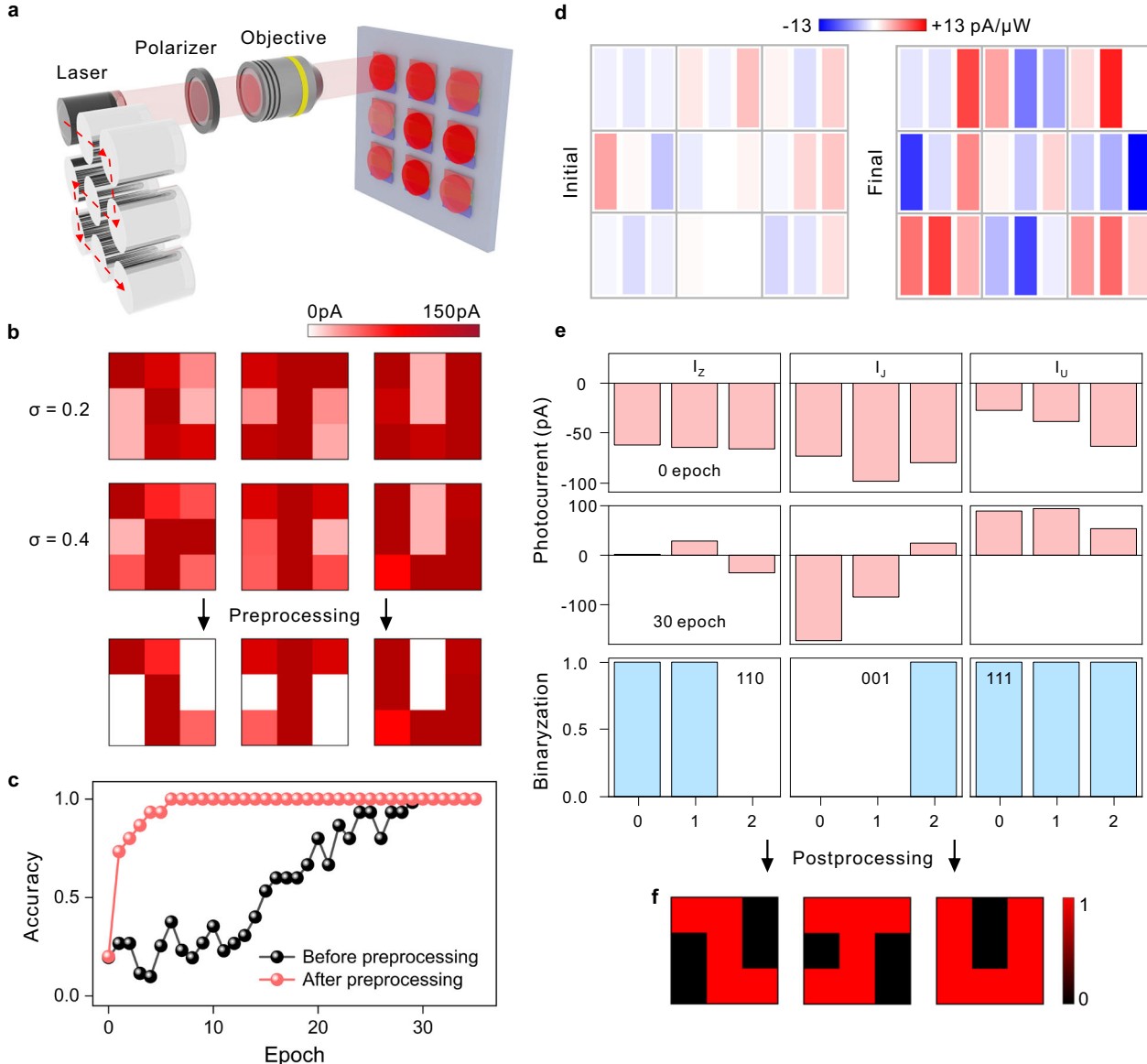

**Fig. 4 | AVPRM operation as a classifier. a** Schematic illustration of the optical setup for network training/operation. The resulting image is projected onto the photodiode array in a point-by-point scanning manner. **b** Examples of images with (σ = 0.4) and without (σ = 0.2, 0.4) pre-processing of the device. **c** Comparison of image recognition rate before and after pre-processing of the device. **d** Responsivity distributions before (initial) and after (final) training. **e** The measured three currents corresponding to 'z', 'j' and 'u' target ports, which are converted by the nonlinearity into binary activation codes. In each experiment, the letters 'z', 'j' and 'u' were projected onto the chip separately. **f** The reconstructed letters after post-processing.

Supplementary Fig. 12. We also performed the photoresponsivity measurement when $V_G = -1\,V$ (Supplementary Fig. 13), and the increase of photoresponsivity in the order of magnitude can be applied to image detection and recognition under weak light. Figure 3f shows the photoresponsivity (weight) of the array after 100 off-line training epochs for the letter 'z' in Fig. 3d. The corresponding weights can be written into the array by modulating the voltage $V_{DS}$, and the subsequently projected image generates corresponding output currents for each subpixel. The training processes of the ANN with experimental photoresponsivity curve are illustrated in Supplementary Fig. 14. Figure 3g shows the total output current of the array after each training epoch. The corresponding photoresponsivity of the array after each training epoch is also presented in Supplementary Fig. 15. The currents clearly separate and stabilize after 100 epochs, with the largest current corresponding to the label of the projected letter. Figure 3h show the transfer characteristic curves of the PPTs obtained under different

incident optical powers at 635 nm wavelength. The transfer characteristic curves of the PPTs obtained under different incident optical powers at 532 and 473 nm wavelengths are also presented in Supplementary Fig. 16. Besides, the transfer characteristic curves of the PPTs corresponding to different optical wavelengths obtained under relatively small drain-source voltage ($V_{DS} = 0.1\,V$) are also shown in Supplementary Fig. 23a–c. The performance summary and detailed analysis of individual phototransistors are provided in Supplementary Table 1 and Supplementary Note 2, respectively. The dynamic range (DR) is defined by the equation: $DR = 20 \times \log[I_{max}/I_{min}]$ (dB), where $I_{max}$ and $I_{min}$ are the photocurrent values corresponding to the maximum and minimum gate voltages, respectively. The calculated effective DR is up to 180 dB, which equals almost the highest value reported up to date[5,11]. The reason and mechanism of the device with ultra-high DR are shown in Supplementary Fig. 22 and Supplementary Note 1. As shown in Fig. 3i, by applying different levels of gate voltage

$V_G$ to each pixel in the array, the image noise is gradually weakened, the image contrast is gradually enhanced, and its main features are eventually fully displayed. Similarly, the features of the image can also be clearly presented under small gate voltage modulation, as shown in Supplementary Fig. 23d–f, although the clarity is generally weaker than that under large gate voltage modulation. Therefore, the characteristic allows us to realize image pre-processing such as contrast enhancement and noise reduction by locally modulating the gate voltage of each pixel.

## Implementations of pattern classification

To test the integrated sensing, pre-processing and image recognition functions of the AVPRM chip, we used it as a classifier to recognize the letters 'z', 'j' and 'u'. For training and testing of the chip, a point-by-point scan is used to project the optical image using the setup shown in Fig. 4a (for details, see Methods). In our current setup, the weights of the ANN are stored in an external memory and delivered to each PPT detector via a cabling. In this example of supervised learning algorithm, cross-entropy is used as the loss/cost function, the weight values were updated by backpropagation of the gradient of the loss function[20]. A detailed flow chart of the whole AVPRM including the training algorithm is presented in Supplementary Fig. 9c. Figure 4b illustrates the input image with different Gaussian noise (σ = 0.2, 0.4) added and the pre-processed image (σ = 0.4), which is extracted from the drain-source current $I_D$. After applying gate voltage $V_G$ to the certain pixel (the white pixels in Fig. 4b), the body feature of the letters in the pre-processed image has been enhanced obviously. The complete dataset used for training after pre-processing is given in Supplementary Fig. 17. In Fig. 4c, the accuracy of recognition with and without pre-processing of the images is plotted. For the pre-processed image, it is faster to reach recognition accuracy of 100%. The initial and final responsivities/weights of the classifier are shown in Fig. 4d, and the measured currents and corresponding codes of the target port for each letter are depicted in Fig. 4e. Each code corresponds to a letter, and the corresponding letter is reconstructed through post-processing, as shown in Fig. 4f. To evaluate the overall performance (processing speed and energy consumption) of this network, we also performed time-resolved measurements. The experimental setup is shown in Supplementary Fig. 9b. The trigger/measurement pulse is provided in Supplementary Fig. 18a (see Method for details). The response of a single spike in a single device measured with the assistance of gate voltage is approximately 500 ns (Supplementary Fig. 18b) and the leakage current is shown in Supplementary Fig. 18c. The dissipated energy per spike of the device with such sensitive photoresponse is approximately $2.4 \times 10^{-17}$ J, according to $W = I \times V \times t$[16]. In order to illustrate the high-speed capabilities of PPTA, we carried out measurements by employing a 500 ns pulsed laser source and an electric pulse source with synchronous triggering. As previously mentioned, the PPTA functioned as a classifier and was pre-trained. We then projected two letters ('z' and 'u') and measured the time-resolved signals of three channels in sequence. As shown in Supplementary Fig. 19a, each pixel contained in the image is illuminated on the PPTA with a pulsed laser at a different power $P_N$. Upon optical stimulation, a total output current $I_M$ is generated by a circuit in the array consisting of all the $M$th subpixels connected in a neural network manner. Subsequently, the generated current $I_M$ is amplified by the preamplifier and converted into voltage $V_M$ input into the oscilloscope. The principle of generating total current $I_M$ is displayed in Supplementary Fig. 19b. As shown in Supplementary Fig. 19c, d, we plot the electric output pulses, with different output codes representing different image types, which demonstrate the correct pattern classification within ~500 ns. Such a system may hence provide great potential for the development of ultrafast and ultralow power machine vision.

## Discussion

We have summarized recent achievements in artificial neuromorphic devices, as shown in Supplementary Table 2. Compared with other works, our AVPRM device is currently the only fully integrated system that can perform the entire steps from image acquisition to data pre-processing/post-processing in a single device. Due to the enhanced contrast of preprocessed image through the device, such an integrated multifunctional AVPRM has shown significant improvement in recognition rate and efficiency for image processing. In addition, due to the compatibility of the manufacturing process with complementary metal oxide semiconductor technology, the device can be presented and operated at an array scale. Thus, this allows the device to be one of the few that can be used for on-site recognition of images after training on it. As a differentiation from previous plasmonic devices, the proof-of-concept device we designed introduces plasmonic nanogratings and utilizes the strong coupling effect to increase light absorption, converting specific wavelength optical signals (RGB) into electrical signals, thus additionally introducing carriers, which greatly increases the energy efficiency of the device. It is precisely because of this design that, under the synergistic effect of gate electrodes and nanogratings, the device has achieved the best comprehensive performance (dynamic range 180 dB, high speed 500 ns, ultralow energy consumption $2.4 \times 10^{-17}$ J per spike) in the existing neuromorphic devices while possessing multiple functions.

Another important question is the ultrafast recognition capability of the device. Although the entire process from the generation of hot electrons by plasmon decay to the injection into MoS₂ is accomplished on a sub-nanosecond level[27,29], the transfer of the hot electrons and the establishment of the thermoelectric potential prolong the entire process. Further solutions can be developed by doping $MoS_2/WSe_2$ respectively, using split-gate electrodes to establish potential differences, thereby assisting rapid migration and detection of hot electrons, ultimately enabling ultrafast image recognition (Supplementary Fig. 20). Considering the future mass production and cost of the device, the simpler the device architecture and the fewer the processing steps, the greater its potential for machine vision applications. From this perspective, the device structure could be simplified while maintaining its main performance, for example, by adopting an on-chip integrated structure or simply by using a few layers $MoS_2$ material as the channel. To demonstrate the feasibility of the latter approach, we present an plasmon-enhanced photodetector in Supplementary Fig. 21. Under the irradiation of different power light, the short-circuit photocurrent is generated under the effect of the thermoelectric potential generated by the plasmon excited by Ag nanograting. Different photoresponsivity can be tuned by modulating the ITO bottom gate electrode and source-drain polarity. This tunable photoresponsivity (weight) can then be applied to the subsequent training of ANN devices and image recognition.

In conclusion, we have presented an AVPRM composed of PPTA, which integrates multifunctions of sensing, pre-processing and image recognition simultaneously. The strong coupling effect caused by the WPPs structure greatly enhances the absorption of the tricolor light in the device, thus improving the generation of hot electrons and the injection into the floating gate. Under the coordination of photo-thermoelectric effect caused by plasmon dephasing and electrical modulation, the current on/off ratio of the device exceeds ~$1 \times 10^9$ and the dynamic range reaches 180 dB. The performance of the device can greatly enhance the image contrast during the pre-processing process. Subsequent image recognition is successfully performed under the incidence of continuous light and pulsed light, respectively. Two letters with a duration of 500 ns can be recognized on the basis of consuming $2.4 \times 10^{-17}$ J per spike. By performing image pre-processing using this PPT, the image quality is effectively improved, and the efficiency and accuracy of subsequent image recognition is increased. This device exhibits great potential in terms of large dynamic range,

ultrafast and ultralow power consumption for machine vision applications.

## Methods

### Device fabrication

The fabrication of the chip follows the procedure described in Supplementary Fig. 2. A quartz wafer was used as the original substrate, which was cleaned with acetone, isopropyl alcohol and deionized water, respectively. The cleaned quartz wafer was deposited with a layer of ITO film (~200 nm) using magnetron sputtering (Denton Discovery-635). Subsequently, an $Al_2O_3$ layer was grew on top of the ITO film by atomic layer deposition (~40 nm, Kurt J. Lesker ALD150LX). 2D crystals including $MoS_2$ (thicknesses: ~8 nm, lateral dimension: ~90 × 60 μm), h-BN (thicknesses: ~10 nm, lateral dimension: ~105 × 80 μm) and $WSe_2$ (thicknesses: ~7 nm, lateral dimension: ~87 × 60 μm) flakes were derived from bulk source materials by a mechanical peel-transfer method. For the transfer of $MoS_2$ flake, it was first mechanically exfoliated on a transparent polydimethylsiloxane film and then transferred to the substrate with the help of an optical microscope. To eliminate unnecessary stresses, the transferred 2D $MoS_2$ was annealed in an argon atmosphere. Standard e-beam lithography (EBL, Raith Voyager) and magnetron sputtering were then employed to define the Ti/Ag nanogratings on the produced structures by a lift-off approach. Next, we defined the mask with EBL and carried out reactive ion etching (RIE) with $Ar/SF_6$ plasma to separate the previously transferred $MoS_2$ sheet into 27 pixels. Afterwards, the mask was removed with acetone. 2D h-BN and $WSe_2$ flakes were also transferred to the structure using the same method described above. In order to maximize the absorption of nanogratings, $Ar/SF_6$ plasma was again used to perform RIE towards 2D heterostructure on the mask defined by EBL. The top metal layer (gate electrode and drain-source electrode) was added by another EBL process and Cr/Au (3 nm/15 nm) evaporation. Finally, $Al_2O_3$ (20 nm) and Cr/Au (20 nm/50 nm) layers were deposited on the produced heterostructures by lift-off methods using standard EBL process and magnetron sputtering/thermal evaporation of materials.

### Experimental setup

Schematics of the experimental setup are shown in Supplementary Figs. 8 and 9a, b. Light from a semiconductor laser (635/532/473 nm wavelength) was collimated by a lens before passing through a linear polarizer. The polarization direction of the linear polarizer was mounted perpendicular to the long axis of the Ag nanograting, and the linearly polarized light was projected on the structure in a normal incidence manner. The gray level of each pixel in the optical image was achieved by adjusting the laser power, and then the optical image was projected onto the sample using a microscope objective with a long working distance. A source meter (Keithley, 2400) was used to supply gate voltage to the PPT, and a source meter (Keithley, 2450) was used to supply drain-source voltage to the PPT while measuring the output current. The sample was connected to the source meter via a home-made measurement box and BNC connection cable. For time-resolved measurements, a femtosecond pulsed laser source (BFL-1030-20B, BWT) was used, which was triggered using a lock-in amplifier (Stanford Research Systems, SR830) to emit a single pulse at a wavelength of 515 nm. The 500 ns cycle drain-source pulse voltage was provided by an arbitrary waveform generator (Keithley, 3390), and the output current was amplified by a preamplifier (Stanford Research Systems, SR570) and converted into a voltage signal, which was finally recorded by an oscilloscope (Siglent). In addition, all measurements were carried out at room temperature in an air environment.

### Simulation and strong coupling model

The transmittance spectra and electromagnetic field distributions of the structures with strong coupling were simulated using finite-difference time-domain method. The plane wave light source was projected onto the structure with normal incidence in the direction of polarization perpendicular to the long axis of Ag nanogratings. In order to highlight the strong coupling effect, we neglected the effect of 2D materials in our experimental and theoretical simulations. Here, small volumes Ag nanorods with a height of 20 nm were selected to form the grating in order to achieve large photoelectric conversion efficiency by reducing the proportion of radiation damping and increasing the ballistic transport probability[27] and hot electron relaxation time[39]. All calculated data were collected while satisfying the steady state energy criteria.

A coupled oscillator model was introduced to analyze the strong coupling behavior of the hybrid architecture under specific parameters. The plasmon of Ag nanogratings, symmetrized photonic mode, and antisymmetrized mode can be assumed as three oscillators. Therefore, the Hamiltonian of this three-coupled system can be written as:

$$\begin{pmatrix} E_{Pl} - i\gamma_{Pl}/2 & g_w & g_s \\ g_w & E_{Asym} - i\gamma_{Asym}/2 & 0 \\ g_s & 0 & E_{Sym} - i\gamma_{Sym}/2 \end{pmatrix} \quad (2)$$

Where $\gamma_{Pl}$, $\gamma_{Asym}$, and $\gamma_{Sym}$ are the linewidths of plasmon, antisymmetrized and symmetrized modes, $E_{Pl}$, $E_{Asym}$, and $E_{Sym}$ are the resonance energies of plasmon, antisymmetrized and symmetrized modes, while $g_w$ and $g_s$ represent plasmon-antisymmetrized mode and plasmon-symmetrized mode interaction constants. In the three-oscillator model, the eigenstates of Hamiltonian correspond to the three hybrid branches. The wave function of each branch from the admixture contribution of plasmon, symmetric mode and antisymmetric mode can be expressed as $|\psi_j\rangle = \alpha_{Pl}^j |Pl\rangle + \alpha_{Sym}^j |Sym\rangle + \alpha_{Asym}^j |Asym\rangle$, where $\alpha_i^j (i = Pl, Sym, Asym; j = Upper, Middle, Lower)$ denotes Hopfield coefficients. The modular square of the Hopfield coefficient represents the proportion of uncoupled states $\mathbf{P}^i = (P_{Upper}^i, P_{Middle}^i, P_{Lower}^i)$ in hybrid state. Also, the weight of each hybrid branch $\mathbf{W} = (W_{Upper}, W_{Middle}, W_{Lower})$ in this strong coupling regime can be calculated as $W_j = \gamma_j / \sum_j \gamma_j$.

### Image recognition task

In our proposed AVPRM, the pattern classification task was solved by a single-layer perceptron containing nine input neurons and one output neuron. The hardware implementation of a single-layer perceptron was accomplished by interconnecting 3 × 3 PPTs in an ANN manner to form a PPTA. The network was trained off-line using computer simulation, a method called the ex-situ training. Subsequently, the pre-determined photoresponsivity matrix, that is, photoresonsivities scaled from dimensionless weights, was transferred to the PPTA to complete the image recognition. The network was trained by MATLAB. The direction of weight update for each training epoch was determined by the positive or negative value of the delta-rule weight increments $\Delta$, where $\Delta = P_n(\phi_m(I) - \phi(I'_m))$ here is exactly delta-rule weight increments. Here, $\phi(I'_m)$ is the training value, $\phi_m(I)$ is the target value and $P_n$ is the incident light power of the $n$th pixel with noise.

## Data availability

Relevant data supporting the key findings of this study are available within the article and the Supplementary Information file. All raw data generated during the current study are available from the corresponding authors upon request.

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

## Acknowledgements

This work was supported by the National Key R&D Program of China (2019YFA0308602), the National Science Foundation of China (general program 12174336 & major program 91950205) and the Natural Science Foundation of Zhejiang Province (LR20A040002). We thank the Micro and Nano Fabrication Centre at Zhejiang University for facility support and W. Wang at the State Key Laboratory of Modern Optical Instru-mentation for suggestions on nanofabrication. We appreciate the equipment support provided by the Center of Electron Microscopy of Zhejiang University for the preparation of samples to be characterized, as well as the assistance provided by H. Huang from the Center for Micro/Nano Fabrication of Westlake University for sample characteriza-tion. We also acknowledge useful comments from Prof. D. Xiang of Frontier Institute of Chip and System, Fudan University.

## Author contributions

L.L. and T.Z. conceived and designed the project. T.Z. designed and built the experimental setup, programmed the machine-learning algorithm, fabricated the ANN PPTA, carried out the material and device char-acterization. X.F., Z.W., Y.T. and D.W. provided assistance with material characterization. T.Z. and L.L. analyzed data and wrote the manuscript. X.G., P.W. and L.T. provided suggestions for data analysis. All authors commented on the manuscript.

## Competing interests

The authors declare no competing interests.
