## [Peer Review File · Nature Communications]

High performance artificial visual perception and recognition with a plasmon-enhanced 2D material neural networkEditorial Note: Supplementary Figure 22 in this Peer Review File has been amended to remove third-party material where no permission to publish could be obtained.

REVIEWER COMMENTS

Reviewer #1 (Remarks to the Author):

In this manuscript, the authors developed an artificial visual system (AVS) consisting of a 3x3 plasmonic phototransistor array (PPTA) as an artificial neural network, which can sense, pre-process and recognize the optical signals. The plasmonic phototransistor consists of 2 parts, an optical sensor and a transistor, which are connected through MoS₂ and a floating gate. The optical sensor utilizes Ag nanograting, a plasmonic nanostructure, to excite plasmon polaritons, and then create hot electrons that can be injected into MoS₂ layer. The thermoelectric potential caused by those hot electrons can modulate the channel current in transistor, enabling signal detection. The PPTAs developed by the authors demonstrate exceptional dynamic range (180 dB), fast response (500 ns) and remarkably low power consumption (2.4×10^{-17} J).

However, the structure of their AVS as well as the neural network training methods bear some similarities with previous works (Nature Communications 9, 5106 (2018); Nature 579, 62–66 (2020); Nature Communications 11, 4602 (2020)), although they applied a different mechanism (plasmonic polaritons) in enhancing optical response. The application of Ag plasmonic nanograting in sensing and imaging has also been reported before (Scientific Report 6, 29984 (2016); Scientific Report 7, 15985 (2017); IEEE Journal of Selected Topics in Quantum Electronics 27 (1), 4600710 (2021)). To differentiate their research, the authors need to emphasize unique aspects and highlight their distinct contributions more explicitly.

Additionally, there are some other issues that authors should address when revising their manuscript:

- In page 2, the authors claimed “For current AVS, a hardware solution with both the pre-processing function of the human retina and the image recognition capability of the visual cortex has not been reported, especially in time-critical applications”. Since the device configuration is very similar to previous works (Nature Communications 9, 5106 (2018); Nature 579, 62–66 (2020)), can authors clarify in more details how their devices differentiate from others regarding to pre-processing and recognition.
- The mechanisms in Fig. 2 need to be improved, particularly with regards to the bending of the MoS₂ energy band, which can cause confusion for readers. Can authors provide the complete band diagram of semiconductor MoS₂ and WSe₂?
- It would be more convincing to quantify the enhancement effect of Ag nanograting if the authors can compared the results from samples without Ag nanograting.
- The authors did not show memory effect of their device. Based on the literature, memory effect plays a crucial role in artificial vision system. The authors should explain how they address this issue in their AVS.
- In the method section, the authors are requested to provide the thicknesses and lateral dimensions of the flacks of 2D materials obtained by mechanical exfoliation.

Reviewer #2 (Remarks to the Author):

T. Zhang et al. reported an artificial visual system with a plasmon-enhanced 2D materials neural network. The plasmonic phototransistor array (PPTA), made up of nanograting and 2D heterostructures, integrates synchronous sensing, preprocessing, and image recognition functions. By harnessing the synergistic effects of hybrid localized surface plasmon resonance (LSPR) and waveguide modes, the authors achieve remarkably efficient photoelectric conversion.

Overall, this work gives an interesting solution for an artificial visual system, which proves its potential with a large dynamic range (180 dB), high speed (50 Ons), and ultra-low power consumption per spike (2.4×10^{-17} J). Therefore, the reviewer will recommend this work accepted in Nature Communications after addressing the following issues.

1. The authors claim high performance artificial visual system in the title and demonstrate image processing in Figure 4. What are the advantages and disadvantages of the proposed device compared with previous works for high-performance image processing?
2. To elucidate the high-speed (500 ns) characteristic more explicitly, the authors could consider incorporating a figure similar to Figure 5 in Nature 579, 62–66 (2020) <https://doi.org/10.1038/s41586-020-2038-x>, which effectively demonstrates the rapid recognition of two distinct letters.
3. Figure 4 demonstrates AVS operation as a classifier. However, the authors haven't clearly shown related device features in Figure 1-3. In other words, the authors may reorganize the figures and add some discussions to connect device features to the application.
4. The working speed of this mechanism should be able to reach the sub-nanosecond level. Please discuss what limits its speed and suggest future device structure designs for achieving faster speeds.
5. Are there particular reasons for the use of 2D material MoS₂ and WSe₂? Using other materials is also workable?
6. The device structure seems relatively complicated. Authors may give a discussion on how to simplify it while keeping the same performance. Is it potential to realize the on-chip integration of electronic-photonic devices?
7. Minor issues: for line 215 "photoresponse is approximately 2.4×10^{-17} J, according to $P = I \times V \times t$ ". The power P equals $I \times V$, while the energy W is $I \times V \times t$.

Reviewer #3 (Remarks to the Author):

Summary:

In this paper, the authors report an AVS composed of PPTA, which constitutes an ANN that integrates simultaneous sensing, pre-processing and image recognition functions. The author increased the efficiency and accuracy of subsequent image recognition by performing image pre-processing using this PPT. I have major and minor comments which do not allow me to recommend the publication in its current form.

Comments:

1. The work spans from the material/device level all the way up to the circuit architecture and algorithm level. However, it is also this broadness that makes judging the work difficult, as the authors combine state-of-the-art technologies to create a more complex system that is difficult to benchmark in its entirety. The authors fail to put the performance of their building blocks into the context of prior art (Nat Commun 9, 5106 (2018), Nature 579, 62–66 (2020) and Nat Commun 11, 5934 (2020)), where clearly bench-marking would be possible and desirable.

2. The authors explain the excellence of 2D materials in the context of their work in the introduction. However, in the main results, there is no content that can be connected to the excellence of 2D materials mentioned in the introduction. What benefits did 2D materials provide for demonstrating the AVS reported by the authors? Can other materials not be considered? Justification for the use of 2D materials is required through experimental tasks and/or comparison with previously reported papers.

3. Characterization of the device is vastly lacking. There are not sufficient information on quality, thickness, scale of materials existing in stacks; although SEM and TEM image are provided, considering broad readership of Nature Communications, it is not sufficient. It is necessary to provide more information on the device. For example, thickness profiles of each materials via AFM, quality of materials via Raman and/or PL analysis, and so on. Each metric can affect the performance of the device.

4. The authors built an array using 2D flakes? Then, what is the uniformity of each device, and how can uniformity affect the efficiency and accuracy of image recognition?

4-1) Just providing transfer characteristics under dark conditions (provided in Extended Data. Fig. 7i) is not sufficient.

4-2) Please provide clear and large image provided in Extended Data Fig. 2t.

4-3) There seems to be a difference in scale information between Extended Data Fig. 2r-s and Extended Data Fig. 4b. What devices were used for the data provided in the main article?

5. It was not easy to understand the connection between the device characterization results provided in Fig.3 and the image recognition results provided in Fig.4. The authors provided a process in extended data Fig.6, but it is very vague. It would be good to provide detailed recognition process with respect to device operation. Also, it is necessary to provide experimental results in the middle stage of image recognition (e.g., device conductivity at each epoch), and detailed conditions in the image recognition task needed to be provided to the method part.

5-1) What are the recognition results under various conditions provided in Fig.3?

6. Isn't learning (backpropagation) using an external computer? Then, is it correct to assert that the AVS reported in this article is an ANN-integrated device? The authors provided a neural network schematic in Figure 1e. This can be misunderstood that the device/array the authors fabricated itself has the learning function.

Reviewer #4 (Remarks to the Author):

The author realized hardware devices connected in an artificial neural network (ANN) that can simultaneously sense, pre-process and recognize optical images without latency by designing PPTA. This work provides a hardware solution for artificial neural vision: a hardware solution that combines the preprocessing function of the human retina and the image recognition ability of the visual cortex. The constructed device exhibits large dynamic range (180 dB), high speed (500 ns) and ultralow energy consumption per spike (2.4×10^{-17} J). However, there are still issues that need to be addressed further.

1. The transfer characteristics of the phototransistor should be given. Also, the gate leakage current should also be measured. The transistor performance is the base for the further study on the artificial visual system, however, the data or discussion in the present manuscript is absent. I do not understand the origin for the high performance mentioned by the authors.
2. The difference of the MoS₂ floating gate and the metal floating gate should be clarified.
3. What is the main contribution of 2D materials? Why the WSe₂ was selected as the channel material?
4. The bridge between transistor and the network operation is also absent. The network training is based on a single device or a hardware transistor network? The author should offer more details on the transistor array if they exist.
5. The definition of response speed, specific calculation methods and figures need to be given.
6. Please explain the reason and mechanism of the ultra-high DR of the device.
7. Figure 1f reflects the real shape and structure of the device, please mark the source, drain, gate and other structures of the device.
8. How to evaluate recognition accuracy, the definition of target current (signal) and training current (signal), etc., please give in SI.
9. The author claims that the device is a hardware solution for artificial vision with both sensing and recognition functions. Please give specific information and corresponding meanings of each step of device training and image recognition in SI to help readers better understanding of the work. The author needs to show readers in detail the specific information of using the constructed device to complete image training and image recognition.

Table of Contents

- Responses to Reviewer #1’s comments: page 1
- Responses to Reviewer #2’s comments: page 12
- Responses to Reviewer #3’s comments: page 24
- Responses to Reviewer #4’s comments: page 39

=====

Responses to Reviewer #1’s comments

=====

[Comment 1-1]

In this manuscript, the authors developed an artificial visual system (AVS) consisting of a 3×3 plasmonic phototransistor array (PPTA) as an artificial neural network, which can sense, pre-process and recognize the optical signals. The plasmonic phototransistor consists of 2 parts, an optical sensor and a transistor, which are connected through MoS_2 and a floating gate. The optical sensor utilizes Ag nanograting, a plasmonic nanostructure, to excite plasmon polaritons, and then create hot electrons that can be injected into MoS_2 layer. The thermoelectric potential caused by those hot electrons can modulate the channel current in transistor, enabling signal detection. The PPTAs developed by the authors demonstrate exceptional dynamic range (180 dB), fast response (500 ns) and remarkably low power consumption (2.4×10^{-17} J).

[Author reply 1-1]

We appreciate the Referee's valuable time to evaluate our manuscript and make an in-depth summary. We are very grateful for the valuable comments from the Referee. In response to the Referee’s concerns as well as various suggestions, we have updated and revised the paper accordingly, which are summarized below in a point-to-point fashion.

[Comment 1-2]

However, the structure of their AVS as well as the neural network training methods bear some similarities with previous works (Nature Communications 9, 5106 (2018); Nature 579, 62–66 (2020); Nature

Communications 11, 4602 (2020)), although they applied a different mechanism (plasmonic polaritons) in enhancing optical response. The application of Ag plasmonic nanograting in sensing and imaging has also been reported before (Scientific Report 6, 29984 (2016); Scientific Report 7, 15985 (2017); IEEE Journal of Selected Topics in Quantum Electronics 27 (1), 4600710 (2021)). To differentiate their research, the authors need to emphasize unique aspects and highlight their distinct contributions more explicitly.

[Author reply 1-2]

We appreciate the professional comments and valuable suggestions from the Referee. According to the Referee's nice comments, we have benefited from additional knowledge about the progress in artificial neuromorphic systems and plasmon-enhanced sensing/imaging devices, so we cited the articles mentioned by the Referee (Refs [39-42], in the updated manuscript).

In response to the Referee’s suggestion, in order to differentiate previous researches and emphasize the unique aspects and distinctive contributions of our work, we have summarized recent achievements in artificial neuromorphic devices, as shown in Table 1. Compared with other works, our AVS device is currently the only fully integrated system that can perform the entire steps from image acquisition to data pre-processing/post-processing in a single device. Due to the enhanced contrast of preprocessed image through the device, such an integrated multifunctional AVS device has shown significant improvement in recognition rate and efficiency for image processing. In addition, due to the compatibility of the manufacturing process with CMOS technology, the device can be presented and operated at an array scale. Thus, this allows the device to be one of the few that can be used for on-site recognition of images after training on it. As a differentiation from previous plasmonic devices, the proof-of-concept device we designed introduces plasmonic nanogratings and utilizes the strong coupling effect to increase light absorption, converting specific wavelength optical signals (RGB) into electrical signals, thus additionally introducing carriers, which greatly increases the energy efficiency of the device. It is precisely because of this design that, under the synergistic effect of gate electrodes and nanogratings, the device has achieved the best comprehensive performance (dynamic range 180 dB, high speed 500 ns, ultralow energy consumption 2.4×10^{-17} J per spike) in the existing neuromorphic devices while possessing multiple functions.

Table 1 Comparison of the proposed neuromorphic device with previous report.

Pixel structure	Integrated function	OSR?	E	t	DR	CC?	Ref.
-----------------	---------------------	------	---	---	----	-----	------

A hBN/WSe ₂ synaptic device + hBN/WSe ₂ photodetector	Sensing + postprocessing (pattern recognition)	N/A	66-532 fJ	10 ms	N/A	Yes	16
A WSe ₂ photodiode	Sensing + postprocessing (classify/encode images)	Yes	N/A	40 ns	N/A	Yes	19
A resistive pressure sensor + perovskite-based photodetector + hydrogel-based ionic cable and a synaptic transistor	Sensing (visual-haptic fusion) + postprocessing (pattern recognition)	N/A	N/A	1 s	N/A	N/A	39
A MoS ₂ -pV3D3 phototransistor	Sensing + preprocessing	N/A	N/A	0.5 s	N/A	Yes	2
A synaptic device with a structure of Pd/MoO _x /ITO	Sensing + preprocessing (background noise reduction)	N/A	N/A	200 ms	N/A	Yes	1
A MoS ₂ phototransistor + UVO treatment	Sensing + preprocessing (Scotopic/Photopic adaptaton)	N/A	N/A	N/A	199 dB	Yes	5
A MoS ₂ /hBN/WSe ₂ plasmonic phototransistor	Sensing + preprocessing (image contrast enhancement) + postprocessing (image recognition)	Yes	2.4×10^{-17} J	500 ns	180 dB	Yes	This work

DR stands for dynamic range and can be used to enhance image contrast (preprocessing) by adjusting

DR. **E**: energy consumption. **t**: response time. **OSR**: On-site recognition. **CC**: CMOS compatibility.

Correspondingly, we modified the following:

1. We cited four new articles (references [39-42]) in the updated manuscript on page 2, lines 34, 42.
2. We added the paragraph "We have summarized recent achievements while possessing multiple functions." to the Discussion section in the updated manuscript on pages 14-15, lines 282-296.
3. We added a new table (Table 1) in the updated manuscript on pages 15-16, lines 297-299.

[Comment 1-3]

Additionally, there are some other issues that authors should address when revising their manuscript:

- *In page 2, the authors claimed "For current AVS, a hardware solution with both the pre-processing function of the human retina and the image recognition capability of the visual cortex has not been reported, especially in time-critical applications". Since the device configuration is very similar to previous works (Nature Communications 9, 5106 (2018); Nature 579, 62–66 (2020)), can authors clarify in more details how their devices differentiate from others regarding to pre-processing and recognition.*

[Author reply 1-3]

We would like to thank the Referee for raising critical questions. We present in Fig. R1 the circuit diagram of a single neuromorphic device from the two references mentioned by the Referee and from our work. The devices in Figs. R1a [*Nature Communications 9, 5106 (2018)*] and R1b [*Nature 579, 62–66 (2020)*] both contain sensing modules (highlighted in red dashed boxes), which utilize the photoresponsive characteristics of the 2D material (WSe₂) itself to achieve sensing functions. The devices also include postprocessing modules (highlighted in blue dashed boxes), which modulate the perceived electrical signals through pulse voltage V_{pulse} [*Nature Communications 9, 5106 (2018)*] or a pair of gate voltages V_G / $-V_G$ [*Nature 579, 62–66 (2020)*] to achieve recognition function. It can be seen that neither of these devices contains a preprocessing module. Unlike the previous two, Fig. R1c presents a neuromorphic device [*Our work*] that incorporates both sensing, pre-processing and postprocessing modules. In the photoelectric conversion process of the sensing module (highlighted in red dashed boxes), in addition to utilizing the photoresponsive properties of the 2D material (MoS₂) itself, an additional hybrid plasmonic structure is introduced to generate hot electrons, which greatly increases the energy efficiency of the device. In the pre-processing module (highlighted in green dashed boxes), a side gate voltage V_G is applied to modulate the transport of carriers in each pixels in the PPTA, thus achieving the effect of enhancing image contrast. After the preprocessed signal is passed to the postprocessing module (highlighted in blue dashed boxes), the drain-source voltage V_{SD} is applied to modulate the weight of each pixel in the PPTA to achieve the goal of

image recognition. Such an integrated multifunctional AVS device has shown significant improvement in recognition rate and efficiency for image processing. In addition, the network is trained off-line using computer simulation. Subsequently, the predetermined photoresponsivity matrix is transferred to the device to complete the image recognition. Taking this and the Referee's comment into account, we have changed the term "time-critical applications" to "on-site critical applications".

Figure R1. Circuit diagram of a single neuromorphic device. **a**, Simplified electrical circuit for the optoneural synaptic device [*Nature Communications* 9, 5106 (2018)]. **b**, Circuit diagram of a single pixel in the photodiode array [*Nature* 579, 62–66 (2020)]. **c**, Circuit diagram of a single pixel in the PPTA [*Our work*].

Correspondingly, we modified the following:

1. We changed the term "time-critical applications" to "on-site critical applications" in the updated manuscript on page 2, line 36.

[Comment 1-4]

The mechanisms in Fig. 2 need to be improved, particularly with regards to the bending of the MoS₂ energy band, which can cause confusion for readers. Can authors provide the complete band diagram of semiconductor MoS₂ and WSe₂?

[Author reply 1-4]

We thank the Referee for this useful comment, and we agree that providing the complete band diagram of semiconductor MoS₂ and WSe₂ can improve the quality and readability of the article. Compared to the

previous figure, we draw the complete MoS₂ band diagram in Fig. 2b. Besides, the band diagram was divided into two parts, which correspond to the Ag/MoS₂ architecture on the left side of the PPT device and the MoS₂/hBN/WSe₂ architecture on the right side. The figure is attached below. As shown in the lower part of Fig. 2e, the hot electrons generated by the decay of plasmons are injected into the conduction band of MoS₂, and then the hot electrons are transported to the right side by tilting the energy band under the action of thermoelectric potential. Subsequently, the electrons transported to the right side of MoS₂ induced holes in the valence band of WSe₂, which were then used for current measurement. Obviously, the higher the optical power, the larger the measured channel current. After the light is turned off (Fig. 2f), the electrons are restored to the initial state by the accumulated potential E_A . When the light is turned on and the negative side gate voltage $-V_G$ is applied, the electrons will be more easily transferred from the left side of MoS₂ to the right side, as there is an additional gate potential E_G (Fig. 2g). From the perspective of energy band (lower part of Fig. 2g), it can be explained that the energy band of MoS₂ is more inclined under the combined effect of thermoelectric potential (E_T) and gate potential (E_G), making it easier for electrons to be transported to the right side. Accordingly, the larger channel current will be induced by the MoS₂ gate. Conversely, by applying a positive gate voltage $+V_G$ while the light is turned on, the electrons will be dragged to the left side because of the additional gate potential (E_G) (Fig. 2h). From the perspective of energy band (lower part of Fig. 2h), in this case, the energy band of MoS₂ are tilted in the opposite direction due to the effect of the gate potential (E_G), which makes it difficult for the electrons to overcome this potential to reach the right side. Naturally, only weak current can be measured in WSe₂ channel.

Fig. 2b Simplified band diagram illustrating the hot electron injection process taking place at the Ag-MoS₂ interface. In addition to receiving hot electrons emitted from the Ag nanograting, MoS₂ itself can also

generate a small amount of electrons after receiving light. **e-h** Charge-flow illustrations and schematic band diagrams at different operation modes: light on (e), light off (f), light on and apply $-V_G$ (g), light on and apply $+V_G$ (h). E_C represents the conduction band, E_V represents the valence band, E_F represents the fermi level and χ represents the electron affinity. The blue balls denote the holes, the magenta balls denote the electrons, and the magenta arrows indicate the flow direction of the electrons. E_T represents the thermoelectric potential, E_A represents the accumulated potential, and E_G represents the gate potential. The black arrows represent the direction of each potential. The black dotted arrow represents the direction of the electron transition.

Correspondingly, we modified the following:

1. We added the sentence "The band diagram shown the accumulated potential E_A ." in the updated manuscript on pages 8-9, lines 159-168.
2. We added the sentence "From the perspective of to the right side." in the updated manuscript on page 9, lines 171-173.
3. We added the sentence "From the perspective of to reach the right side." in the updated manuscript on page 9, lines 176-178.
4. We revised a figure (Fig. 2) in the updated manuscript on page 6, line 113.

[Comment 1-5]

It would be more convincing to quantify the enhancement effect of Ag nanograting if the authors can compare the results from samples without Ag nanograting.

[Author reply 1-5]

We thank the Referee for this useful comment, and we agree that it would be more convincing to provide a quantitative comparison of the photocurrent of devices with and without nanogratings. As shown in Supplementary Fig. 6, we present the photocurrent of 2D PPT with and without Ag nanogratings. It can be seen from Supplementary Fig. 6c that only subtle change in the photocurrent can be detected in the WSe₂ channel when the laser is turned on and off, indicating that only a small amount of photogenerated carriers are generated in the MoS₂ floating gate in devices without Ag nanogratings. When the Ag nanograting is integrated into the PPT device, as shown in Supplementary Fig. 6d, significant changes of photocurrent can

be detected in the device when the laser is turned on and off. This indicates that a large number of electrons are generated after the decay of plasmons excited by the Ag nanograting. Then, this portion of electrons is transported to the right side of MoS₂ under the large photothermoelectric potential caused by plasmon decay, thereby inducing sufficient holes in the WSe₂ channel for current detection.

Supplementary Figure 6: Photocurrent of 2D PPT with and without Ag nanogratings. **c**, When the laser is turned on and off, only subtle changes of photocurrent can be detected in the WSe₂ channel, indicating that very few electrons in the MoS₂ excited by the device without Ag nanograting. **d**, When the laser is turned on and off, a significant change of photocurrent is detected in the WSe₂ channel, indicating that a large number of electrons are excited through plasmon decay in devices with Ag nanogratings. The generated electrons are transferred to the right side of the MoS₂ floating gate under the photothermoelectric effect and induce a large number of holes in the WSe₂ channel.

Correspondingly, we modified the following:

1. We added the sentence "The quantitative comparison of transport of hot electrons (see Supplementary Fig. 6)." in the updated manuscript on page 7, lines 135-137.
2. We revised a figure (Supplementary Fig. 6) in the updated Supplementary Information on page 7.

[Comment 1-6]

The authors did not show memory effect of their device. Based on the literature, memory effect plays a crucial role in artificial vision system. The authors should explain how they address this issue in their AVS.

[Author reply 1-6]

We thank the Referee for the comment. In response, as shown in Fig. R2, we have summarized several commonly used architectures for neuromorphic computing. As shown in Fig. R2a, the traditional visual architecture includes physically separated sensors, memory, and processing units, where the shuttle of a large amount of raw and unstructured data between each unit brings notable energy consumption issues, time delays, and hardware costs [*Nature* 579, 32-33 (2020)]. To overcome this, computing architectures will need to shift from a computing-centric approach to a data-centric approach, where some of the computational tasks need to be moved to sensory devices at the outer edges of the computer system, reducing unnecessary data movement. Based on this goal, various disruptive architectures have been proposed to eliminate or reduce data transfer and conversion at sensor, memory, and processor interfaces.

With the emergence of memristive materials, in-memory computing techniques that integrate memory and computing modules have been widely developed. As shown in Fig. R2b, this architecture blurs the boundary between memory and processing units by folding them into one module, while the sensor array remains unchanged [*Nature Materials* DOI 10.1038/s41563-023-01676-0 (2023)]. In this architecture, the weights are stored in local memory and the computation tasks occur simultaneously. However, naturally non-volatile materials, such as memristive materials and ferroelectrics, do not meet the requirements for high-performance photoelectric detection. This will limit the operating speed of the device and increase inefficient power consumption. An alternative approach, as shown in Fig. R2c, is near-sensor and in-sensor computing [*Nature Electronics* 3, 664-671 (2020)]. In a near-sensor computing paradigm, processing units or accelerators are positioned in close proximity to the sensors. This spatial arrangement facilitates the localized execution of specific computational tasks at the sensor endpoints, resulting in an enhanced sensor-to-processor interface. Consequently, this design minimizes the necessity to transfer redundant data over extended distances. In the architecture of in-sensor computing, individual sensors that are self-adaptive or networks of interconnected sensors possess the capability to autonomously carry out the direct processing of sensory information. This integrated functionality serves to eliminate the need for a distinct sensor-to-processor interface, effectively merging the tasks of sensing and computing. This architectural framework operates without the need for a memory unit and aims to provide effective sensory data processing. For example, Mennel and co-workers directly implemented an artificial neural network in their image sensor, which can simultaneously sense and process optical images without latency and without any memory units [*Nature* 579, 62-66 (2020)]. The PPTA architecture discussed in our work belongs to the in-sensor computing architecture, where ANN weights are stored in an external memory and supplied to each PPT detector via cabling. It is worth mentioning that, as shown in Fig. R2d, a new in-memory sensing and computing (IMSC) architecture has recently emerged, which integrates sensing, weight memorization and

high-level computing functions. On the other hand, there is still room for improvement in ultra-fast time response and preprocessing functions [Nature Materials DOI 10.1038/s41563-023-01676-0 (2023)].

a Conventional computing architecture

b In-memory computing architecture

c In/Near-sensor computing architecture

d In-memory sensing and computing architecture

Figure R2. Different sensory computing architectures. **a**, Conventional sensory computing architecture with separate sensing, memory and computing units. **b**, In-memory computing architecture with folded memory and computing units. **c**, In-sensor computing architecture and Near-sensor computing architecture. In in-sensor computing architecture, the processing functions are embedded in the sensors for front-end processing. The sensors can collaborate together to perform information processing, data aggregation and

compression, eliminating data transmission between the sensors and processors. In near-sensor computing architecture, individual sensors are connected to front-end processing units through advanced integrated circuit packaging technologies for real-time readout and processing. The front-end near-sensor processing units implement a portion of the processing tasks. **d**, In-memory sensing and computing (IMSC) architecture. The IMSC architecture integrates image sensing, weight memorization, and computing modules.

Correspondingly, we modified the following:

1. We added the sentence "In our current setup, the weights of the ANN are stored in an external memory and delivered to each PPT detector via a cabling." in the updated manuscript on page 12, lines 245-246.

[Comment 1-7]

In the method section, the authors are requested to provide the thicknesses and lateral dimensions of the flakes of 2D materials obtained by mechanical exfoliation.

[Author reply 1-7]

We thank the Referee for pointing this issue and we have added the thicknesses and lateral dimensions of the flakes of 2D materials.

Correspondingly, we modified the following:

1. We added the thicknesses and lateral dimensions of the flakes of 2D materials in red font in the updated manuscript on page 17, lines 327-329.

=====

Responses to Reviewer #2's comments

=====

[Comment 2-1]

T. Zhang et al. reported an artificial visual system with a plasmon-enhanced 2D materials neural network. The plasmonic phototransistor array (PPTA), made up of nanograting and 2D heterostructures, integrates synchronous sensing, preprocessing, and image recognition functions. By harnessing the synergistic effects of hybrid localized surface plasmon resonance (LSPR) and waveguide modes, the authors achieve remarkably efficient photoelectric conversion.

Overall, this work gives an interesting solution for an artificial visual system, which proves its potential with a large dynamic range (180 dB), high speed (500 ns), and ultra-low power consumption per spike (2.4×10^{-17} J). Therefore, the reviewer will recommend this work accepted in Nature Communications after addressing the following issues.

[Author reply 2-1]

We gratefully thank the Referee for the expert reviewing and recommending the acceptance of our work. In particularity, we thank the Referee for pointing out that “Overall, this work gives an interesting solution for an artificial visual system”. We have addressed the Referee’s concerns by revising the paper accordingly, fully taking his/her comments into considerations. The changes are summarized below in a point-to-point fashion.

[Comment 2-2]

1. The authors claim high performance artificial visual system in the title and demonstrate image processing in Figure 4. What are the advantages and disadvantages of the proposed device compared with previous works for high-performance image processing?

[Author reply 2-2]

We thank the Referee for the professional comment and valuable questions raised. In response to the Referee’s comment, we have summarized recent achievements in artificial neuromorphic devices, as shown

in Table 1. Compared with other works, our AVS device is currently the only fully integrated system that can perform the entire steps from image acquisition to data pre-processing/post-processing in a single device. Due to the enhanced contrast of preprocessed image through the device, such an integrated multifunctional AVS device has shown significant improvement in recognition rate and efficiency for image processing. In addition, due to the compatibility of the manufacturing process with CMOS technology, the device can be presented and operated at an array scale. Thus, this allows the device to be one of the few that can be used for on-site recognition of images after training on it. As a differentiation from previous plasmonic devices, the proof-of-concept device we designed introduces plasmonic nanogratings and utilizes the strong coupling effect to increase light absorption, converting specific wavelength optical signals (RGB) into electrical signals, thus additionally introducing carriers, which greatly increases the energy efficiency of the device. It is precisely because of this design that, under the synergistic effect of gate electrodes and nanogratings, the device has achieved the best comprehensive performance (dynamic range 180 dB, high speed 500 ns, ultralow energy consumption 2.4×10^{-17} J per spike) in the existing neuromorphic devices while possessing multiple functions.

The relative complexity of our device architecture compared to current top-level devices limits the carrier transport speed to sub-nanosecond levels, which is a disadvantage of the device. The device improvement for this disadvantage will be discussed in detail below.

Table 1 Comparison of the proposed neuromorphic device with previous report.

Pixel structure	Integrated function	OSR?	E	t	DR	CC?	Ref.
A hBN/WSe ₂ synaptic device + hBN/WSe ₂ photodetector	Sensing + postprocessing (pattern recognition)	N/A	66-532 fJ	10 ms	N/A	Yes	16
A WSe ₂ photodiode	Sensing + postprocessing (classify/encode images)	Yes	N/A	40 ns	N/A	Yes	19
A resistive pressure sensor + perovskite-based photodetector	Sensing (visual-haptic fusion) +	N/A	N/A	1 s	N/A	N/A	39

+ hydrogel-based ionic cable and a synaptic transistor	postprocessing (pattern recognition)						
A MoS ₂ -pV3D3 phototransistor	Sensing + preprocessing	N/A	N/A	0.5 s	N/A	Yes	2
A synaptic device with a structure of Pd/MoO _x /ITO	Sensing + preprocessing (background noise reduction)	N/A	N/A	200 ms	N/A	Yes	1
A MoS ₂ phototransistor + UVO treatment	Sensing + preprocessing (Scotopic/Photopic adaptaton)	N/A	N/A	N/A	199 dB	Yes	5
A MoS ₂ /hBN/WSe ₂ plasmonic phototransistor	Sensing + preprocessing (image contrast enhancement) + postprocessing (image recognition)	Yes	2.4×10^{-17} J	500 ns	180 dB	Yes	This work

DR stands for dynamic range and can be used to enhance image contrast (preprocessing) by adjusting

DR. **E**: energy consumption. **t**: response time. **OSR**: On-site recognition. **CC**: CMOS compatibility.

Correspondingly, we modified the following:

1. We added the paragraph "We have summarized recent achievements while possessing multiple functions." to the Discussion section in the updated manuscript on pages 14-15, lines 282-296.
2. We added a new table (Table 1) in the updated manuscript on pages 15-16, lines 297-299.

[Comment 2-3]

3. To elucidate the high-speed (500 ns) characteristic more explicitly, the authors could consider incorporating a figure similar to Figure 5 in Nature 579, 62–66 (2020) <https://doi.org/10.1038/s41586-020-2038-x>, which effectively demonstrates the rapid recognition of two distinct letters.

[Author reply 2-3]

We thank the Referee for the professional comment and useful suggestion. In response, we have added a graph similar to Figure 5 in Nature (*Nature* 579, 62-66 (2020)) to effectively demonstrate rapid recognition of two different letters. The figure is also attached below. In order to illustrate the high-speed capabilities of PPTA, we carried out measurements by employing a 500-ns pulsed laser source and an electric pulse source with synchronous triggering. As previously mentioned, the PPTA functioned as a classifier and was pre-trained. Then, we projected two letters ('z' and 'u') and measured the time-resolved signals of three channels in sequence. As shown in Supplementary Fig. 19, we plot the electric output pulses, with different output codes representing different image types, which demonstrate the correct pattern classification within ~500 ns. Such a system may hence provide great potential for the development of ultrafast machine vision.

Supplementary Figure 19: Ultrafast image recognition. Projection of two different letters, 'z' and 'u', with a duration of 500 ns, leads to distinct output voltage codes 110 and 111 for the three channels. CH represents the output channel. The encoding rule for output voltage are the same as in the maintext, with positive signals marked as 1 and negative signals marked as 0.

Correspondingly, we modified the following:

1. We added the sentence "In order to illustrate the pattern classification within ~500 ns." in the updated manuscript on page 14, lines 273-279.
2. We added a new figure (Supplementary Fig. 19) in the updated Supplementary Information on page 18.

[Comment 2-4]

3. Figure 4 demonstrates AVS operation as a classifier. However, the authors haven't clearly shown related device features in Figure 1-3. In other words, the authors may reorganize the figures and add some discussions to connect device features to the application.

[Author reply 2-4]

We appreciate the professional comments and valuable suggestions from the Referee. In order to show the related device features more clearly, we have reorganized Fig. 3 and added some discussions. The figure is also attached below. Here we choose the red light of $\lambda = 635$ nm, and its power (0-10 μ W) is divided into 11 orders. Figure 3a presents the multi-state photocurrents corresponding to different levels of optical power. These photocurrents are graphically visualised as 11 grey levels in the 0-1 interval. Thus, the gray level of each pixel in the image can be extracted and presented through photocurrent measurement, as shown in Fig. 3b. By measuring the photocurrent corresponding to three wavelengths of light at the same power $P = 10$ μ W, we can distinguish red (635 nm), green (532 nm) and blue colors (473 nm) when $V_{DS} = 0.1$ V (Fig. 3c). As shown in Fig. 3d, when the photocurrent of the pixel with the largest gray level in the image is measured, the color of the image can be distinguished by different current values.

Considering the subsequent ANN training, we plotted the voltage tunable photocurrents corresponding to each grey level, as shown in Fig. 3e. Figure 3f shows the photoresponsivity (weight) of the array after 100 off-line training epochs for the letter 'z' in Fig. 3d. The corresponding weights can be written into the array by modulating the voltage V_{DS} , and the subsequently projected image generates corresponding output currents for each subpixel. The training processes of the ANN with experimental photoresponsivity curve are illustrated in Supplementary Fig. 14. The figure is also attached below. Figure 3g shows the total output current of the array after each training epoch. The corresponding photoresponsivity of the array after each training epoch is also presented in Supplementary Fig. 15. The figure is also attached below. The currents clearly separate and stabilize after 100 epochs, with the largest current corresponding to the label of the projected letter. Figure 3h show the transfer characteristic curves of the PPTs obtained under different incident optical powers at 635nm wavelength. As shown in Fig. 3i, by applying different levels of gate voltage V_G to each pixel in the array, the image noise is gradually weakened, the image contrast is gradually enhanced, and its main features are eventually fully displayed. Therefore, the characteristic allows us to realize image pre-processing such as contrast enhancement and noise reduction by locally modulating the gate voltage of each pixel.

Fig.3 Functional implementation of 2D PPTA. **a**, Multi-state photocurrents corresponding to different levels of optical power (grey levels), where the laser wavelength is 635 nm and the drain-source voltage is 0.1 V. **b**, Extraction of the gray level of each pixel in the image. The images were added with Gaussian noise (with standard deviation of $\sigma = 0.3$). **c**, Photocurrent of different colors of light (R: 635nm, G: 532nm, B: 473nm) under the same measurement conditions, where the power is 10 μW and the drain-source voltage is 0.1 V. **d**, Recognition of different color images. The numbers in different colored boxes represent the maximum gray level extracted from (b) and its corresponding current. **e**, The voltage (V_{DS}) tunable photocurrent corresponding to each gray scale. **f**, The photoresponsivity of the array and its modulation process after 100 off-line training epochs of the letter 'z' in (d). **g**, The total output current of the projected letter after each training epoch, where the letter 'z' in (d) is projected onto the chip. The maximum current corresponds to the label of the projected letter. **h**, The transfer characteristic curves of the devices with red light measured under different P values at $V_{DS} = 1$ V, respectively. **i**, The pre-processing process of images. The left column represents the image with Gaussian noise ($\sigma = 0.4$) added before pre-processing, the middle

column represents the modulation voltage required for pre-processing, and the right column represents the image after pre-processing.

Supplementary Figure 14: The training processes of the ANN with experimental photoresponsivity curve. The ANN is trained off-line according to the experimental photoresponsivity curve in Supplementary Fig. 10c. R_+ represents the positive potentiation part of the photoresponsivity curve, while R_- represents the negative depression part of the photoresponsivity curve.

Supplementary Figure 15: Photoresponsivity and weight distributions of the array. a-d, The corresponding photoresponsivity of the array after 0 (a), 10(b), 30(c), and 50(d) training epochs, respectively. e-f, Photoresponsivity (e) and weight (f) distributions before (initial) and after (final) training.

Correspondingly, we modified the following:

1. We added a few sentences in red font in the updated manuscript on pages 9-12.
2. We revised a figure (Fig. 3) in the updated manuscript on page 10.
3. We added two new figure (Supplementary Fig. 14 and Supplementary Fig. 15) in the updated Supplementary Information on pages 15-16.

[Comment 2-5]

4. The working speed of this mechanism should be able to reach the sub-nanosecond level. Please discuss what limits its speed and suggest future device structure designs for achieving faster speeds.

[Author reply 2-5]

We thank the Referee for the insightful comment and valuable suggestion. The ultra-fast recognition capability of neuromorphic devices is an important indicator for machine vision applications. Although the entire process from the generation of hot electrons by plasmon decay to the injection into MoS₂ is accomplished on a sub-nanosecond level, the transfer of the hot electrons and the establishment of the thermoelectric potential prolong the entire process. Further solutions can be developed by doping MoS₂/WSe₂ respectively, using split-gate electrodes to establish potential differences, thereby assisting rapid migration and detection of hot electrons, ultimately enabling ultrafast image recognition. We show in Supplementary Fig. 20 the reasons and further solution regarding the speed limitation of the device operation. The figure is also attached below.

Supplementary Figure 20: Discussion on the limited speed of device operation. **a**, Schematic of the PPT. The bottom of the figure shows the energy band diagram of MoS₂ under light on and light off conditions, respectively. Under the condition of light on, the thermoelectric potential generated by the plasmon effect tilts the energy band, leading to the transport of hot electrons from the hot end to the cold end. The establishment of the thermoelectric potential and the transport of hot electrons prolongs the whole process. **b**, Schematic of the improved PPT. In this device, the bottom ITO and the top shielding layer are made into splitting electrodes respectively. As shown in the energy band diagram at the bottom, MoS₂/WSe₂ energy band can be modulated into PN junction energy band after applying positive/negative gate voltage to the splitting electrode. The hot electrons generated by plasmon decay and the induced carriers in the channel will migrate rapidly under the effect of the energy potential, thus accelerating the whole process.

Correspondingly, we modified the following:

1. We added the sentence "Another important question is enabling ultrafast image recognition." to the Discussion section in the updated manuscript on page 16, lines 300-305.
2. We added a new figure (Supplementary Fig. 20) in the updated Supplementary Information on page 19.

[Comment 2-6]

5. Are there particular reasons for the use of 2D material MoS₂ and WSe₂? Using other materials is also workable?

[Author reply 2-6]

We use 2D MoS₂ as a floating gate layer mainly based on the following considerations: **(1) 2D thin layered structure:** The increase in the energy efficiency (such as light absorption efficiency, electron injection rate) of the device comes from the strong coupling effect of the LSPR mode in the Ag nanograting and the waveguide mode in the ITO. In order to avoid weakening the strong coupling effect, it is imperative to minimize the thickness of the material between the Ag nanograting and the ITO waveguide. Therefore, it is appropriate to choose a MoS₂ with 2D thinness sandwiched between the two layers. **(2) n-type semiconductor:** The image recognition function of the PPTA device is achieved based on the injection and elimination of electrons in the MoS₂ floating gate. MoS₂ is an n-type semiconductor that exhibits high electron mobility, making it ideal for electron injection. **(3) Large Seebeck coefficient:** Local absorption of light enhanced by plasmons produces local heating on the left side of MoS₂ layer in the device. This local heating creates a temperature gradient ΔT between the left and right sides of MoS₂. According to $\Delta V_{\text{PTE}} = S \times \Delta T$, when the temperature gradient ΔT is determined, the photothermoelectric voltage ΔV_{PTE} is directly proportional to the Seebeck coefficient S , while the Seebeck coefficient of MoS₂ is several orders of magnitude higher than other materials such as graphene [*Nano Letters* 13, 358-363 (2013)]. **(4) External tunability:** 2D semiconductors, rather than conventional materials with fixed chemical doping distributions, provide the possibility of external tunability of potential distribution in devices (*Nature* 579, 62-66 (2020)). The preprocessing function of the device inspired by the human eye is mainly realized through the electrical external regulation of MoS₂. By applying a gate voltage to the MoS₂ layer, the electron concentration and its transport direction can be controlled, thereby modulating the potential distribution to adjust the gray level of each pixel. **(5) Environmental stability:** The PPTA device we demonstrate are designed and fabricated for future practical applications in machine vision, so it is necessary to use 2D materials with high stability. Unlike some other 2D materials, MoS₂ is relatively stable in ambient conditions, which is important for practical device applications. Therefore, we chose MoS₂ as the floating gate.

We use 2D WSe₂ as the channel layer mainly based on the following considerations: **(1) p-type semiconductor:** The trained weight (photoresponsivity) of the PPT device is written by adjusting the drain-source voltage in the channel conduction state. The conduction state of the WSe₂ channel is achieved by inducing sufficient holes in the WSe₂ layer through electrons in the lower MoS₂ layer. WSe₂ is a p-type semiconductor with high electron mobility, making it an ideal carrier for holes. **(2) High carrier mobility:** WSe₂ exhibits high carrier mobility, which is a measure of how quickly charge carriers (electrons and holes) can move through the material (*Science* 344, 725-728 (2014)). This is the foundation for PPT device to achieve high-speed image recognition. **(3) Environmental stability:** The PPTA device we demonstrate are designed and fabricated for future practical applications in machine vision, so it is necessary to use 2D

materials with high stability. Similar to 2D MoS₂ material, WSe₂ is relatively stable in ambient conditions, which is important for practical device applications. Therefore, we chose WSe₂ as the channel material.

[Comment 2-7]

6. The device structure seems relatively complicated. Authors may give a discussion on how to simplify it while keeping the same performance. Is it potential to realize the on-chip integration of electronic-photonic devices?

[Author reply 2-7]

We appreciate the useful comment and suggestion from the Referee. Considering the future mass production and cost of the device, the simpler the device architecture and the fewer the processing steps, the greater its potential for machine vision applications. From this perspective, the device structure could be simplified while maintaining its main performance, for example, by adopting an on-chip integrated structure or simply by using a few layers MoS₂ material as the channel. To demonstrate the feasibility of the latter approach, we present an plasmon-enhanced photodetector in Supplementary Fig. 21. Under the irradiation of different power light, the short-circuit photocurrent is generated under the effect of the thermoelectric potential generated by the plasmon excited by Ag nanograting. Different photoresponsivity can be tuned by modulating the ITO bottom gate electrode and source-drain polarity. This tunable photoresponsivity (weight) can then be applied to the subsequent training of ANN devices and image recognition.

Supplementary Figure 21: Plasmon-enhanced photodetector with adjustable photoresponsivity (weight). **a**, Schematic of the plasmon-enhanced photodetector. The device operates under short-circuit conditions and sets the photoresponsiveness by supplying a voltage to the ITO bottom gate electrode against V_G . The device is operated under short-circuit conditions and the photoresponsivity is set by supplying a voltage V_G to the ITO bottom gate electrode. The thickness of each material in the device is consistent with

the maintext. **b**, The short-circuit photocurrent measured by modulating the bottom ITO gate voltage under different optical power, where the positive/negative photocurrents were obtained by switching the polarity of the source/drain controlled by an external circuit. **c**, Voltage tunability of the photoresponsivity extracted from (b). The weight (photoresponsivity) of the ANN training process can be obtained by adjusting the gate voltage.

Correspondingly, we modified the following:

1. We added the sentence "Considering the future mass production of ANN devices and image recognition." to the Discussion section in the updated manuscript on pages 16-17, lines 305-315.
2. We added a new figure (Supplementary Fig. 21) in the updated Supplementary Information on page 20.

[Comment 2-8]

7. *The device Minor issues: for line 215 "photoresponse is approximately 2.4×10^{-17} J, according to $P = I \times V \times t$ ". The power P equals $I \times V$, while the energy W is $I \times V \times t$.*

[Author reply 2-8]

We thank the Referee for pointing out this issue we have changed the formula to $W = I \times V \times t$.

Correspondingly, we modified the following:

1. We modified the formula to $W = I \times V \times t$ in the updated manuscript on page 14, line 273.

=====

Responses to Reviewer #3's comments

=====

[Comment 3-1]

Summary:

In this paper, the authors report an AVS composed of PPTA, which constitutes an ANN that integrates simultaneous sensing, pre-processing and image recognition functions. The author increased the efficiency and accuracy of subsequent image recognition by performing image pre-processing using this PPT. I have major and minor comments which do not allow me to recommend the publication in its current form.

[Author reply 3-1]

We thank the Referee for the summary and critical comment on our work. We have addressed the Referee's concerns by revising the paper accordingly, fully taking his/her comments into considerations. The changes are summarized below in a point-to-point fashion.

[Comment 3-2]

Comments:

1. The work spans from the material/device level all the way up to the circuit architecture and algorithm level. However, it is also this broadness that makes judging the work difficult, as the authors combine state-of-the-art technologies to create a more complex system that is difficult to benchmark in its entirety. The authors fail to put the performance of their building blocks into the context of prior art (Nat Commun 9, 5106 (2018), Nature 579, 62–66 (2020) and Nat Commun 11, 5934 (2020)), where clearly bench-marking would be possible and desirable.

[Author reply 3-2]

We thank the Referee for these critical and professional comments. In response to the Referee's concern, we summarized the recent achievements of artificial neuromorphic devices, as shown in Table 1, and comprehensively measured the performance of our building blocks in the context of prior art (Nat Commun 9, 5106 (2018), Nature 579, 62–66 (2020) and Nat Commun 11, 5934 (2020)). Compared with other works, our AVS device is currently the only fully integrated system that can perform the entire steps from image

acquisition to data pre-processing/post-processing in a single device. Due to the enhanced contrast of preprocessed image through the device, such an integrated multifunctional AVS device has shown significant improvement in recognition rate and efficiency for image processing. In addition, due to the compatibility of the manufacturing process with CMOS technology, the device can be presented and operated at an array scale. Thus, this allows the device to be one of the few that can be used for on-site recognition of images after training on it. As a differentiation from previous plasmonic devices, the proof-of-concept device we designed introduces plasmonic nanogratings and utilizes the strong coupling effect to increase light absorption, converting specific wavelength optical signals (RGB) into electrical signals, thus additionally introducing carriers, which greatly increases the energy efficiency of the device. It is precisely because of this design that, under the synergistic effect of gate electrodes and nanogratings, the device has achieved the best comprehensive performance (dynamic range 180 dB, high speed 500 ns, ultralow energy consumption 2.4×10^{-17} J per spike) in the existing neuromorphic devices while possessing multiple functions.

Table 1 Comparison of the proposed neuromorphic device with previous report.

Pixel structure	Integrated function	OSR?	E	t	DR	CC?	Ref.
A hBN/WSe ₂ synaptic device + hBN/WSe ₂ photodetector	Sensing + postprocessing (pattern recognition)	N/A	66-532 fJ	10 ms	N/A	Yes	16
A WSe ₂ photodiode	Sensing + postprocessing (classify/encode images)	Yes	N/A	40 ns	N/A	Yes	19
A resistive pressure sensor + perovskite-based photodetector + hydrogel-based ionic cable and a synaptic transistor	Sensing (visual-haptic fusion) + postprocessing (pattern recognition)	N/A	N/A	1 s	N/A	N/A	39
A MoS ₂ -pV3D3 phototransistor	Sensing + preprocessing	N/A	N/A	0.5 s	N/A	Yes	2

A synaptic device with a structure of Pd/MoO _x /ITO	Sensing + preprocessing (background noise reduction)	N/A	N/A	200 ms	N/A	Yes	1
A MoS ₂ phototransistor + UVO treatment	Sensing + preprocessing (Scotopic/Photopic adaptaton)	N/A	N/A	N/A	199 dB	Yes	5
A MoS ₂ /hBN/WSe ₂ plasmonic phototransistor	Sensing + preprocessing (image contrast enhancement) + postprocessing (image recognition)	Yes	2.4×10^{-17} J	500 ns	180 dB	Yes	This work

DR stands for dynamic range and can be used to enhance image contrast (preprocessing) by adjusting

DR. **E**: energy consumption. **t**: response time. **OSR**: On-site recognition. **CC**: CMOS compatibility.

Correspondingly, we modified the following:

1. We added the paragraph "We have summarized recent achievements while possessing multiple functions." to the Discussion section in the updated manuscript on pages 14-15, lines 282-296.
2. We added a new table (Table 1) in the updated manuscript on pages 15-16, lines 297-299.

[Comment 3-3]

2. The authors explain the excellence of 2D materials in the context of their work in the introduction. However, in the main results, there is no content that can be connected to the excellence of 2D materials mentioned in the introduction. What benefits did 2D materials provide for demonstrating the AVS reported by the authors? Can other materials not be considered? Justification for the use of 2D materials is required through experimental tasks and/or comparison with previously reported papers.

[Author reply 3-3]

We thank the Referee for raising the questions. Some excellent properties of 2D materials are mentioned in the introduction of the maintext, such as ultrafast response, external tunability and large

photothermoelectric effect. The ultrafast response characteristics of 2D materials indicate that charge carriers can pass through them quickly. This is essential for high-performance electronic and optoelectronic devices. We performed experimental measurements to confirm the high-speed image recognition capability (Supplementary Fig. 19) of PPT devices based on 2D material stacking. Therefore, we mentioned the ultrafast response properties of 2D materials in the introduction. Two dimensional materials exhibit a remarkable sensitivity to external influences, such as electric fields, mechanical strain, and chemical interactions, owing to their ultrathin nature, which can be externally manipulated (*Nature nanotechnology* 12, 901-906 (2017)). Inspired by the human eye, the pre-processing function of the device in the maintext is mainly achieved through the electrical external adjustment of MoS₂. Therefore, we mentioned the external tunable properties of 2D materials in the introduction. The photothermoelectric effect in 2D materials refers to their ability to convert light (photons) into both electrical and thermal energy simultaneously (*Advanced materials* 31, 1902044 (2019)). When these materials are exposed to light, the absorbed photons generate charge carriers (electrons and holes) and, at the same time, induce a temperature gradient due to the heating effect. It is precisely under the effect of this temperature gradient that the carriers of MoS₂ layer of the device flow from the hot end to the cold end, resulting in the channel of the device being in a conductive state. Therefore, we mentioned the large photothermoelectric effect properties of 2D materials in the introduction.

The specific reasons why 2D materials are used in the device are as follows. We use 2D MoS₂ as a floating gate layer mainly based on the following considerations: **(1) 2D thin layered structure:** The increase in the energy efficiency (such as light absorption efficiency, electron injection rate) of the device comes from the strong coupling effect of the LSPR mode in the Ag nanograting and the waveguide mode in the ITO. In order to avoid weakening the strong coupling effect, it is imperative to minimize the thickness of the material between the Ag nanograting and the ITO waveguide. Therefore, it is appropriate to choose a MoS₂ with 2D thinness sandwiched between the two layers. **(2) n-type semiconductor:** The image recognition function of the PPTA device is achieved based on the injection and elimination of electrons in the MoS₂ floating gate. MoS₂ is an n-type semiconductor that exhibits high electron mobility, making it ideal for electron injection. **(3) Large Seebeck coefficient:** Local absorption of light enhanced by plasmons produces local heating on the left side of MoS₂ layer in the device. This local heating creates a temperature gradient ΔT between the left and right sides of MoS₂. According to $\Delta V_{\text{PTE}} = S \times \Delta T$, when the temperature gradient ΔT is determined, the photothermoelectric voltage ΔV_{PTE} is directly proportional to the Seebeck coefficient S , while the Seebeck coefficient of MoS₂ is several orders of magnitude higher than other materials such as graphene [*Nano Letters* 13, 358-363 (2013)]. **(4) External tunability:** 2D semiconductors, rather than conventional materials with fixed chemical doping distributions, provide the possibility of external tunability of potential distribution in devices (*Nature* 579, 62-66 (2020)). The preprocessing function of the device inspired by the human eye is mainly realized through the electrical external regulation

of MoS₂. By applying a gate voltage to the MoS₂ layer, the electron concentration and its transport direction can be controlled, thereby modulating the potential distribution to adjust the gray level of each pixel. **(5) Environmental stability:** The PPTA device we demonstrate are designed and fabricated for future practical applications in machine vision, so it is necessary to use 2D materials with high stability. Unlike some other 2D materials, MoS₂ is relatively stable in ambient conditions, which is important for practical device applications. Therefore, we chose MoS₂ as the floating gate. The h-BN layer, with a large bandgap of 5.97 eV, is used as a high-quality insulator and gate dielectric barrier.

We use 2D WSe₂ as the channel layer mainly based on the following considerations: **(1) p-type semiconductor:** The trained weight (photoresponsivity) of the PPT device is written by adjusting the drain-source voltage in the channel conduction state. The conduction state of the WSe₂ channel is achieved by inducing sufficient holes in the WSe₂ layer through electrons in the lower MoS₂ layer. WSe₂ is a p-type semiconductor with high electron mobility, making it an ideal carrier for holes. **(2) High carrier mobility:** WSe₂ exhibits high carrier mobility, which is a measure of how quickly charge carriers (electrons and holes) can move through the material (*Science* 344, 725-728 (2014)). This is the foundation for PPT device to achieve high-speed image recognition. **(3) Environmental stability:** The PPTA device we demonstrate are designed and fabricated for future practical applications in machine vision, so it is necessary to use 2D materials with high stability. Similar to 2D MoS₂ material, WSe₂ is relatively stable in ambient conditions, which is important for practical device applications. Therefore, we chose WSe₂ as the channel material.

[Comment 3-4]

3. Characterization of the device is vastly lacking. There are not sufficient information on quality, thickness, scale of materials existing in stacks; although SEM and TEM image are provided, considering broad readership of Nature Communications, it is not sufficient. It is necessary to provide more information on the device. For example, thickness profiles of each materials via AFM, quality of materials via Raman and/or PL analysis, and so on. Each metric can affect the performance of the device.

[Author reply 3-4]

We appreciate the useful comment and suggestion from the Referee. The additional analysis on MoS₂, hBN and WSe₂ flakes is described in Supplementary Fig. 5. The figure is also attached below.

Supplementary Figure 5: Characterization of MoS₂, h-BN and WSe₂. **a**, hBN thin film characterized by atomic force microscopy (AFM). **b**, The thickness of hBN thin film measured by AFM along the white line marked in (a). **c**, Raman spectrum on the hBN region. **d**, MoS₂ thin film characterized by AFM. **e**, The thickness of MoS₂ thin film measured by AFM along the white line marked in (d). **f**, Raman spectrum on the MoS₂ region. **g**, WSe₂ thin film characterized by AFM. **h**, The thickness of WSe₂ thin film measured by AFM along the white line marked in (g). **i**, Raman spectrum on the WSe₂ region.

Correspondingly, we modified the following:

1. We added a new figure (Supplementary Figure 5) in the updated Supplementary Information on page 6.
2. We added the sentence "The additional analysis on the MoS₂, h-BN and WSe₂ flakes is described in

Supplementary Fig. 5." in the updated manuscript on page 6, lines 111-112.

[Comment 3-5]

4. The authors built an array using 2D flakes? Then, what is the uniformity of each device, and how can uniformity affect the efficiency and accuracy of image recognition?

4-1) Just providing transfer characteristics under dark conditions (provided in Extended Data Fig. 7i) is not sufficient.

4-2) Please provide clear and large image provided in Extended Data Fig. 2t.

4-3) There seems to be a difference in scale information between Extended Data Fig. 2r-s and Extended Data Fig. 4b. What devices were used for the data provided in the main article?

[Author reply 3-5]

Yes, we built an array using 2D flakes. As shown in Supplementary Fig. 11, the measurement of transmission spectra of 27 PPTs indicates that the device has good uniformity. Due to the good uniformity of the device, weight adjustment can be quickly and effectively achieved, which can help reduce the power consumption of the device and greatly improve the efficiency and accuracy of image recognition. The clear and large image provided in Extended Data Figure 2t is shown in Supplementary Fig. 3. The figure is also attached below.

Yes, there is indeed a difference in the scale information between Extended Data Fig. 2r-s and Extended Data Fig. 4b, as these are displays of two different devices. The data provided in the main article is based on the device shown in Extended Data Figure 2r-s (now Supplementary Figure 2).

Supplementary Figure 3: Full SEM view of the PPTA. Scale bar, 200 μm .

Supplementary Figure 11: PPTA uniformity. The measurement of transmission spectra of 27 PPTs indicates that the device has good uniformity.

Correspondingly, we modified the following:

1. We added the sentence "Also, the measurement device has good uniformity." in the updated manuscript on page 11, lines 211-212.
2. We added two new figures (Supplementary Fig. 3 and Supplementary Fig. 11) in the updated Supplementary Information on page 4 and page 12.

[Comment 3-6]

5. It was not easy to understand the connection between the device characterization results provided in Fig.3 and the image recognition results provided in Fig.4. The authors provided a process in extended data Fig.6, but it is very vague. It would be good to provide detailed recognition process with respect to device operation. Also, it is necessary to provide experimental results in the middle stage of image recognition (e.g., device conductivity at each epoch), and detailed conditions in the image recognition task needed to be provided to the method part.

5-1) What are the recognition results under various conditions provided in Fig.3?

[Author reply 3-6]

We would like to thank the Referee for raising critical questions and valuable suggestions. In order to make the connection between the device characterization results provided in Fig. 3 and the image recognition results provided in Fig. 4 easier to understand, we have updated Fig. 3. The figure is also attached below. Here we choose the red light of $\lambda = 635$ nm, and its power (0-10 μ W) is divided into 11 orders. Figure 3a presents the multi-state photocurrents corresponding to different levels of optical power. These photocurrents are graphically visualised as 11 grey levels in the 0-1 interval. Thus, the gray level of each pixel in the image can be extracted and presented through photocurrent measurement, as shown in Fig. 3b. By measuring the photocurrent corresponding to three wavelengths of light at the same power $P = 10$ μ W, we can distinguish red (635 nm), green (532 nm) and blue colors (473 nm) when $V_{DS} = 0.1$ V (Fig. 3c). As shown in Fig. 3d, when the photocurrent of the pixel with the largest gray level in the image is measured, the color of the image can be distinguished by different current values.

Fig.3 Functional implementation of 2D PPT. **a**, Multi-state photocurrents corresponding to different levels of optical power (grey levels), where the laser wavelength is 635 nm and the drain-source voltage is 0.1 V. **b**, Extraction of the gray level of each pixel in the image. The images were added with Gaussian noise (with standard deviation of $\sigma = 0.3$). **c**, Photocurrent of different colors of light (R: 635nm, G: 532nm, B: 473nm) under the same measurement conditions, where the power is 10 μW and the drain-source voltage is 0.1 V. **d**, Recognition of different color images. The numbers in different colored boxes represent the maximum gray level extracted from (b) and its corresponding current. **e**, The voltage (V_{DS}) tunable photocurrent corresponding to each gray scale. **f**, The photoresponsivity of the array and its modulation process after 100 off-line training epochs of the letter 'z' in (d). **g**, The total output current of the projected letter after each training epoch, where the letter 'z' in (d) is projected onto the chip. The maximum current corresponds to the label of the projected letter. **h**, The transfer characteristic curves of the devices with red light measured under different P values at $V_{DS} = 1$ V, respectively. **i**, The pre-processing process of images. The left column represents the image with Gaussian noise ($\sigma = 0.4$) added before pre-processing, the middle

column represents the modulation voltage required for pre-processing, and the right column represents the image after pre-processing.

Considering the subsequent ANN training, we plotted the voltage tunable photocurrents corresponding to each grey level, as shown in Fig. 3e. Figure 3f shows the photoresponsivity (weight) of the array after 100 off-line training epochs for the letter 'z' in Fig. 3d. The corresponding weights can be written into the array by modulating the voltage V_{DS} , and the subsequently projected image generates corresponding output currents for each subpixel. The training processes of the ANN with experimental photoresponsivity curve are illustrated in Supplementary Fig. 14. The figure is also attached below. Figure 3g shows the total output current of the array after each training epoch. The corresponding photoresponsivity of the array after each training epoch is also presented in Supplementary Fig. 15. The figure is also attached below. The currents clearly separate and stabilize after 100 epochs, with the largest current corresponding to the label of the projected letter. Figure 3h show the transfer characteristic curves of the PPTs obtained under different incident optical powers at 635nm wavelength. As shown in Fig. 3i, by applying different levels of gate voltage V_G to each pixel in the array, the image noise is gradually weakened, the image contrast is gradually enhanced, and its main features are eventually fully displayed. Therefore, the characteristic allows us to realize image pre-processing such as contrast enhancement and noise reduction by locally modulating the gate voltage of each pixel.

The detailed conditions provided in the method section for the image recognition task are as follows: In our proposed AVS, the pattern classification task was solved by a single-layer perceptron containing nine input neurons and one output neuron. The hardware implementation of a single-layer perceptron was accomplished by interconnecting 3×3 PPTs in an ANN manner to form a PPTA. The network was trained off-line using computer simulation, a method called the ex-situ training. Subsequently, the predetermined photoresponsivity matrix, that is, photoresponsivities scaled from dimensionless weights, was transferred to the PPTA to complete the image recognition. The network was trained by MATLAB. The direction of weight update for each training epoch was determined by the positive or negative value of the delta-rule weight increments Δ , where $\Delta = P_n (\phi_m(I) - \phi(I'_m))$ here is exactly delta-rule weight increments. Here, $\phi(I'_m)$ is the training value, $\phi_m(I)$ is the target value and P_n is the incident light power of the n th pixel with noise.

Supplementary Figure 14: The training processes of the ANN with experimental photoresponsivity curve. The ANN is trained off-line according to the experimental photoresponsivity curve in Supplementary Fig. 10c. R_+ represents the positive potentiation part of the photoresponsivity curve, while R_- represents the negative depression part of the photoresponsivity curve.

Supplementary Figure 15: Photoresponsivity and weight distributions of the array. **a-d**, The corresponding photoresponsivity of the array after 0 (**a**), 10(**b**), 30(**c**), and 50(**d**) training epochs, respectively. **e-f**, Photoresponsivity (**e**) and weight (**f**) distributions before (initial) and after (final) training.

Correspondingly, we modified the following:

1. We added a few sentences in red font in the updated manuscript on pages 9-12.
2. We revised a figure (Fig. 3) in the updated manuscript on page 10.
3. We added two new figure (Supplementary Fig. 14 and Supplementary Fig. 15) in the updated Supplementary Information on pages 15-16.
4. We added an “Image recognition task” in the Methods section in the updated manuscript on page 20, lines 384-394.

[Comment 3-7]

6. Isn't learning (backpropagation) using an external computer? Then, is it correct to assert that the AVS reported in this article is an ANN-integrated device? The authors provided a neural network schematic in Figure 1e. This can be misunderstood that the device/array the authors fabricated itself has the learning function.

[Author reply 3-7]

We would like to thank the Referee for raising critical questions. In our proposed AVS, the pattern classification task is solved by a single-layer perceptron containing nine input neurons and one output neuron. The hardware implementation of a single-layer perceptron is accomplished by interconnecting 3×3 PPTs in an ANN manner to form a PPTA. The network is trained off-line using computer simulation, a method called the ex-situ training. Subsequently, the predetermined photoresponsivity matrix, that is, photoresponsivities scaled from dimensionless weights, is transferred to the PPTA to complete the image recognition. The signal at the input end of the network is a preprocessed perceptual signal. Therefore, we wrote the sentence “Herein, we present a PPTA constructed of nanogratings and 2D heterostructures, which constitutes an ANN that integrates simultaneous sensing, pre-processing and image recognition functions.” in the manuscript.

In response to the Referee's concerns and to avoid being misunderstood by readers, we have revised Fig. 1e in the updated manuscript. The figure is also attached below. Figure 1e depicts the entire operation process of AVS in the form of a flowchart. The input optical image is first sensed by the hybrid plasmonic structure in PPT, and the perceived electrical signal is modulated by the side gate electrode in PPT to achieve the pre-processing of the signal. Then, the preprocessed signals are transported to ANN, and the network is trained off-line using computer simulation. Subsequently, the predetermined photoresponsivity

matrix, that is, photoresonivities scaled from dimensionless weights, is transferred to the PPTA to complete the image recognition.

Fig. 1e Illustration of an AVS based on the 2D PPT for image pre-processing and an ANN for image recognition.

Correspondingly, we modified the following:

1. We added the sentence "Figure 1e depicts the entire operation to complete the image recognition." in the updated manuscript on page 5, lines 89-95.
2. We added a new figure (Fig. 1e) in the updated manuscript on page 4, line75.

=====
Responses to Reviewer #4's comments
=====

[Comment 4-1]

The author realized hardware devices connected in an artificial neural network (ANN) that can simultaneously sense, pre-process and recognize optical images without latency by designing PPTA. This work provides a hardware solution for artificial neural vision: a hardware solution that combines the preprocessing function of the human retina and the image recognition ability of the visual cortex. The constructed device exhibits large dynamic range (180 dB), high speed (500 ns) and ultralow energy consumption per spike (2.4×10^{-17} J). However, there are still issues that need to be addressed further..

[Author reply 4-1]

We appreciate the Referee's valuable time in reviewing our manuscript and providing an expert summary. In response to the Referee's concerns as well as various suggestions, we have updated and revised the paper accordingly, which are summarized below in a point-to-point fashion.

[Comment 4-2]

1. The transfer characteristics of the phototransistor should be given. Also, the gate leakage current should also be measured. The transistor performance is the base for the further study on the artificial visual system, however, the data or discussion in the present manuscript is absent. I do not understand the origin for the high performance mentioned by the authors.

[Author reply 4-2]

We would like to thank the Referee for raising critical questions. We present in Fig. R1 the circuit diagram of a single neuromorphic device from the two references and from our work. The devices include postprocessing modules (highlighted in blue dashed boxes), which modulate the perceived electrical signals through pulse voltage V_{pulse} [*Nature Communications* 9, 5106 (2018)] or a pair of gate voltages $V_G/-V_G$ [*Nature* 579, 62–66 (2020)] to achieve recognition function. It can be seen that the photoresponsivity (weight) of these two devices is modulated by the gate voltage in their respective transistors. Therefore, for

these two devices, it is necessary to measure the transfer characteristics. Fig. R1c presents a neuromorphic device [*Our work*] that incorporates both sensing, pre-processing and postprocessing modules. Unlike the above two devices, the photoresponsivity (weight) of our device is modulated by drain-source voltage V_{DS} , so transfer characteristics are not necessary during image recognition. The transfer characteristic curves required during the pre-processing process were measured and presented in the manuscript. By applying different levels of gate voltage V_G to each pixel in the array, the image noise is gradually weakened, the image contrast is gradually enhanced, and its main features are eventually fully displayed. The figure is also attached below, as shown in Fig. R3. The application of gate voltage not only increases the pre-processing function of the device, but also enables the device to achieve an ultra-high dynamic range, as explained in [**Author reply 4-7**].

Figure R3. Transfer characteristic curves. a-c, The transfer characteristic curves of the devices with red (a), green (b) and blue (c) light measured under different P values at $V_{DS} = 1 \text{ V}$, respectively. d, Leakage current when applying side gate voltage.

Besides, the devices in Figs. R1a [*Nature Communications* 9, 5106 (2018)] and R1b [*Nature* 579, 62–66 (2020)] both contain sensing modules (highlighted in red dashed boxes), which utilize the photoresponsive characteristics of the 2D material (WSe₂) itself to achieve sensing functions. In our devices, in the photoelectric conversion process of the sensing module (highlighted in red dashed boxes), in addition to utilizing the photoresponsive properties of the 2D material (MoS₂) itself, an additional hybrid plasmonic structure is introduced to generate hot electrons, which greatly increases the energy efficiency of the device. In the pre-processing module (highlighted in green dashed boxes), a side gate voltage V_G is applied to modulate the transport of carriers in each pixels in the PPTA, thus achieving the effect of enhancing image contrast. After the preprocessed signal is passed to the postprocessing module (highlighted in blue dashed boxes), the drain-source voltage V_{SD} is applied to modulate the weight of each pixel in the PPTA to achieve the goal of image recognition. By performing image pre-processing using this PPT, the image quality is effectively improved, and the efficiency and accuracy of subsequent image recognition is increased.

Figure R1. Circuit diagram of a single neuromorphic device. a, Simplified electrical circuit for the opti-neuromorphic synaptic device [*Nature Communications* 9, 5106 (2018)]. **b,** Circuit diagram of a single pixel in the photodiode array [*Nature* 579, 62–66 (2020)]. **c,** Circuit diagram of a single pixel in the PPTA [*Our work*].

[Comment 4-3]

2. The difference of the MoS₂ floating gate and the metal floating gate should be clarified.

[Author reply 4-3]

We thank the Referee for the professional comment. The main functions of MoS₂ floating gate in the device are: (1) Due to its thin 2D characteristics and unique energy bands, the generation and injection of hot

electrons are efficiently promoted without the strong coupling of the WPPs structure being affected. (2) Under the modulation of gate voltage, 2D MoS₂ with large Seebeck coefficient and high mobility can be used as carrier, which can efficiently assist electrons to complete the transport from left side to right side. The electrons transported to the right side of the MoS₂ floating gate induce holes in the WSe₂ channel located above it, ultimately leading to channel conduction in the device. The conductance of WSe₂ channel in the on-state can be modulated through the regulation of the drain-source voltage V_{DS} , thus realizing the weight regulation of the array. In addition, by changing the polarity of the gate voltage V_G , electrons can be dragged to the left side, causing the channel to be in a off state, thereby achieving brightness adjustment for specific pixels and ultimately achieving the function of image pre-processing. It is obvious that the above effects are difficult to achieve if the MoS₂ floating gate is replaced by a metal floating gate. The main functions of metal floating gate (Au shielding layer) in the device are: (1) To avoid unnecessary direct photocurrents in the WSe₂ channel, the right side is covered by the Al₂O₃/Au layer. (2) In informal measurements, this floating gate is used to test whether the channel is conducting and the electrical connection status. In formal measurements, this metal floating gate is not used for electrical measurements, it is only used as a shielding layer, as mentioned in the manuscript.

[Comment 4-4]

3. What is the main contribution of 2D materials? Why the WSe2 was selected as the channel material?

[Author reply 4-4]

The main contributions of the 2D materials used in the device are as follows. We use 2D MoS₂ as a floating gate layer mainly based on the following considerations: (1) 2D thin layered structure: The increase in the energy efficiency (such as light absorption efficiency, electron injection rate) of the device comes from the strong coupling effect of the LSPR mode in the Ag nanograting and the waveguide mode in the ITO. In order to avoid weakening the strong coupling effect, it is imperative to minimize the thickness of the material between the Ag nanograting and the ITO waveguide. Therefore, it is appropriate to choose a MoS₂ with 2D thinness sandwiched between the two layers. (2) n-type semiconductor: The image recognition function of the PPTA device is achieved based on the injection and elimination of electrons in the MoS₂ floating gate. MoS₂ is an n-type semiconductor that exhibits high electron mobility, making it ideal for electron injection. (3) Large Seebeck coefficient: Local absorption of light enhanced by plasmons produces local heating on the left side of MoS₂ layer in the device. This local heating creates a temperature gradient ΔT between the left and right sides of MoS₂. According to $\Delta V_{PTE} = S \times \Delta T$, when the temperature gradient

ΔT is determined, the photothermoelectric voltage ΔV_{PTE} is directly proportional to the Seebeck coefficient S , while the Seebeck coefficient of MoS_2 is several orders of magnitude higher than other materials such as graphene [*Nano Letters* 13, 358-363 (2013)]. (4) External tunability: 2D semiconductors, rather than conventional materials with fixed chemical doping distributions, provide the possibility of external tunability of potential distribution in devices (*Nature* 579, 62-66 (2020)). The preprocessing function of the device inspired by the human eye is mainly realized through the electrical external regulation of MoS_2 . By applying a gate voltage to the MoS_2 layer, the electron concentration and its transport direction can be controlled, thereby modulating the potential distribution to adjust the gray level of each pixel. (5) Environmental stability: The PPTA device we demonstrate are designed and fabricated for future practical applications in machine vision, so it is necessary to use 2D materials with high stability. Unlike some other 2D materials, MoS_2 is relatively stable in ambient conditions, which is important for practical device applications. Therefore, we chose MoS_2 as the floating gate. The h-BN layer, with a large bandgap of 5.97 eV, is used as a high-quality insulator and gate dielectric barrier.

We use 2D WSe_2 as the channel layer mainly based on the following considerations: (1) p-type semiconductor: The trained weight (photoresponsivity) of the PPT device is written by adjusting the drain-source voltage in the channel conduction state. The conduction state of the WSe_2 channel is achieved by inducing sufficient holes in the WSe_2 layer through electrons in the lower MoS_2 layer. WSe_2 is a p-type semiconductor with high electron mobility, making it an ideal carrier for holes. (2) High carrier mobility: WSe_2 exhibits high carrier mobility, which is a measure of how quickly charge carriers (electrons and holes) can move through the material (*Science* 344, 725-728 (2014)). This is the foundation for PPT device to achieve high-speed image recognition. (3) Environmental stability: The PPTA device we demonstrate are designed and fabricated for future practical applications in machine vision, so it is necessary to use 2D materials with high stability. Similar to 2D MoS_2 material, WSe_2 is relatively stable in ambient conditions, which is important for practical device applications. Therefore, we chose WSe_2 as the channel material.

[Comment 4-5]

4. The bridge between transistor and the network operation is also absent. The network training is based on a single device or a hardware transistor network? The author should offer more details on the transistor array if they exist.

[Author reply 4-5]

We would like to thank the Referee for raising critical questions. The network training is based on a hardware transistor network. To make the connection between individual transistor and transistor array easier to understand, we updated Fig. 3. The figure is also attached below. Here we choose the red light of $\lambda = 635$ nm, and its power (0-10 μW) is divided into 11 orders. Figure 3a presents the multi-state photocurrents corresponding to different levels of optical power. These photocurrents are graphically visualised as 11 grey levels in the 0-1 interval. Thus, the gray level of each pixel (individual transistor) in the image can be extracted and presented through photocurrent measurement, as shown in Fig. 3b. These measurements are specific to individual transistor. By measuring the photocurrent corresponding to three wavelengths of light at the same power $P = 10 \mu\text{W}$, we can distinguish red (635 nm), green (532 nm) and blue colors (473 nm) when $V_{\text{DS}} = 0.1$ V (Fig. 3c). As shown in Fig. 3d, when the photocurrent of the pixel with the largest gray level in the image is measured, the color of the image can be distinguished by different current values. These measurements are also specific to individual transistor.

Fig.3 Functional implementation of 2D PPTA. **a**, Multi-state photocurrents corresponding to different levels of optical power (grey levels), where the laser wavelength is 635 nm and the drain-source voltage is 0.1 V. **b**, Extraction of the gray level of each pixel in the image. The images were added with Gaussian noise (with standard deviation of $\sigma = 0.3$). **c**, Photocurrent of different colors of light (R: 635nm, G: 532nm, B: 473nm) under the same measurement conditions, where the power is 10 μ W and the drain-source voltage is 0.1 V. **d**, Recognition of different color images. The numbers in different colored boxes represent the maximum gray level extracted from (b) and its corresponding current. **e**, The voltage (V_{DS}) tunable photocurrent corresponding to each gray level. **f**, The photoresponsivity of the array and its modulation process after 100 off-line training epochs of the letter ‘z’ in (d). **g**, The total output current of the projected letter after each training epoch, where the letter ‘z’ in (d) is projected onto the chip. The maximum current corresponds to the label of the projected letter. **h**, The transfer characteristic curves of the devices with red light measured under different P values at $V_{DS} = 1$ V, respectively. **i**, The pre-processing process of images. The left column represents the image with Gaussian noise ($\sigma = 0.4$) added before pre-processing, the middle column represents the modulation voltage required for pre-processing, and the right column represents the image after pre-processing.

Considering the subsequent ANN training, we plotted the voltage tunable photocurrents corresponding to each grey level, as shown in Fig. 3e. Figure 3f shows the photoresponsivity (weight) of the array after 100 off-line training epochs for the letter ‘z’ in Fig. 3d. The corresponding weights can be written into the array by modulating the voltage V_{DS} , and the subsequently projected image generates corresponding output currents for each subpixel. Figure 3g shows the total output current of the array after each training epoch. Due to the electrical connection of the array based on Kirchhoff’s law (Fig. 1c), the total current is the same as the result obtained from equation (1) in the main text. The currents clearly separate and stabilize after 100 epochs, with the largest current corresponding to the label of the projected letter. These measurements are specific to transistor network.

Figure 3h show the transfer characteristic curves of the PPTs obtained under different incident optical powers at 635nm wavelength. As shown in Fig. 3i, by applying different levels of gate voltage V_G to each pixel in the array, the image noise is gradually weakened, the image contrast is gradually enhanced, and its main features are eventually fully displayed. Therefore, the characteristic allows us to realize image pre-processing such as contrast enhancement and noise reduction by locally modulating the gate voltage of each pixel. These measurements are specific to individual transistor.

Correspondingly, we modified the following:

1. We added a few sentences in red font in the updated manuscript on pages 9-12.

2. We revised a figure (Fig. 3) in the updated manuscript on page 10.

[Comment 4-6]

5. The definition of response speed, specific calculation methods and figures need to be given.

[Author reply 4-6]

We thank the Referee for the comment. In response, we have added a graph to effectively demonstrate rapid recognition of two different letters. The figure is also attached below. In order to illustrate the high-speed capabilities of PPTA, we carried out measurements by employing a 500-ns pulsed laser source and an electric pulse source with synchronous triggering. As previously mentioned, the PPTA functioned as a classifier and was pre-trained. Then, we projected two letters ('z' and 'u') and measured the time-resolved signals of three channels in sequence. As shown in Supplementary Fig. 19, we plot the electric output pulses, with different output codes representing different image types, which demonstrate the correct pattern classification within ~500 ns. Here, we define the fastest time at which a device can detect a pattern and classify it correctly as the response speed. The specific calculation method for response time is to measure the entire width of the output pulse from the rising edge to the falling edge. From the above, it can be seen that the response speed of PPTA to detect two projected letters and output two significantly different output voltage codes is 500ns.

Supplementary Figure 19: Ultrafast image recognition. Projection of two different letters, 'z' and 'u', with a duration of 500 ns, leads to distinct output voltage codes 110 and 111 for the three channels. CH represents the output channel. The encoding rule for output voltage are the same as in the maintext, with

positive signals marked as 1 and negative signals marked as 0.

Correspondingly, we modified the following:

1. We added the sentence " In order to illustrate the pattern classification within ~500 ns." in the updated manuscript on page 14, lines 273-279.
2. We added a new figure (Supplementary Fig. 19) in the updated Supplementary Information on page 18.

[Comment 4-7]

6. Please explain the reason and mechanism of the ultra-high DR of the device.

[Author reply 4-7]

We thank the Referee for the insightful comment and valuable suggestion. The explanation of the reason and mechanism behind the device's ultra-high DR can help readers better understand our work. According to the Referee's nice comments, we have added Supplementary Fig. 22 to elaborate in detail on the reason and mechanism behind the device's ultra-high DR. The figure is also attached below. For the device without nanograting, as shown in Supplementary Fig. 22a, carriers are generated under light due to the photoresponsive properties of the 2D MoS₂ itself. After applying the gate voltage $-V_G$, a few carriers are dragged to the right end after undergoing a series of processes (e.g., electron-electron scattering, electron-phonon coupling), making it almost difficult for such a small number of carriers to induce charges in the WSe₂ channel. According to $\mu_e = e\tau/m^*$, where e is the elementary charge, τ is the relaxation time, and m^* is the effective mass, it is clear that nothing in this process can cause the carrier mobility μ_e to be increased. For the PPT device with nanograting, as shown in Supplementary Fig. 22b, in addition to the carriers generated under illumination due to the photoresponsive properties of the MoS₂ itself, there is also a portion of hot electrons generated by plasmon decay. In summary, the carrier concentration in the MoS₂ floating gate increases compared to the case in Supplementary Fig. 22a.

On the other hand, the dephasing of plasmon can also cause an increase in electron temperature and lattice temperature. The relaxation time of hot electrons is proportional to ΔT , ΔT is the temperature increase caused by the pump laser [*Chemical Reviews* 111, 3858-3887 (2011)]. Therefore, the mobility of electrons is improved due to the increase in temperature. If the electron mobility can be compared to the acceleration ability of a car, as shown in the lower part of Supplementary Fig. 22, then the difference in electron mobility between the two structures is similar to the difference in acceleration ability between an ordinary car and a

supercar. In addition, the increase in lattice temperature also causes a temperature difference between the two ends of MoS₂, which leads to the formation of thermoelectric potential. After applying the gate voltage $-V_G$, the carriers would be transported toward the right end of MoS₂ under the joint drag of the gate potential E_G and the thermoelectric potential E_T . As a result, the electron concentration (n), mobility (μ_e), and drag electric field ($E_G + E_T$) are greatly enhanced in devices with nanograting, which leads to a significant increase in electron transport to the right end of MoS₂. The accumulation of electrons on the right side of MoS₂ induces a sufficient charge in the WSe₂ channel, and the maximum current (I_{\max}) can be measured by modulating the gate voltage and laser power. On the contrary, as shown in Supplementary Fig. 22c, when a positive gate voltage V_G is applied to generate a potential E_G much greater than the thermoelectric potential E_T , almost no electrons can be transported to the right side of MoS₂ floating gate, and naturally almost no charge can be induced in the WSe₂ channel. In this case, the minimum current (I_{\min}) can be measured. Here, the introduction of nanogratings into the PPT device increases the difference between I_{\max} and I_{\min} , and according to $DR = 20 \times \log[I_{\max} / I_{\min}]$ (dB), ultra-high DR can be obtained.

Supplementary Figure 22: Schematic of the mechanism of the 2D PPT device with ultra-high DR. a, Schematic analysis of electron concentration (n), mobility (μ_e), and drag potential (E_G) in device without nanograting. In this case, only a small number of electrons are dragged to the right side of the MoS₂ floating gate, hardly inducing a charge in WSe₂ channel, which makes the current difficult to be measured. The

lower part of the graph visualizes the mobility (μ_e) as the acceleration ability of an ordinary car. **b**, Schematic analysis of electron concentration (n), mobility (μ_e), and drag potential ($E_G + E_T$) in the PPT device with nanograting. In this case, a sufficient amount of electrons are dragged to the right side of the MoS₂ floating gate, which induces enough charge in WSe₂ channel, leading to a maximum current (I_{\max}) that can be measured. The lower part of the graph visualizes the mobility (μ_e) as the acceleration ability of a supercar. **c**, Schematic of the PPT device in which electrons cannot be dragged to the right side of MoS₂ floating gate under positive gate potential modulation. In this case, almost no electrons can be dragged to the right side of the MoS₂ floating gate, which also cannot induce charges in the WSe₂ channel, resulting in a measurable minimum current (I_{\min}).

Correspondingly, we modified the following:

1. We added the paragraphs "For the device without nanograting in Supplementary Fig. 22a" and "On the other hand ultra-high DR can be obtained." in the updated Supplementary Information on pages 22-23.
2. We added a new figure (Supplementary Fig. 22) in the updated Supplementary Information on page 21.

[Comment 4-8]

7. *Figure 1f reflects the real shape and structure of the device, please mark the source, drain, gate and other structures of the device.*

[Author reply 4-8]

We thank the Referee for this useful comment and we have marked each electrode and structure in the device in Fig. 1f. The figure is also attached below. The blue dashed box shows the drain electrode, the yellow dashed box shows the gate electrode, the red dashed box shows the source electrode, and the green dashed box shows the nanograting.

Fig. 1f Scanning electron microscopy (SEM) image of the PPTA. Scale bar, 20 μm . GND, ground electrode. The blue dashed box shows the drain electrode, the yellow dashed box shows the gate electrode, the red dashed box shows the source electrode, and the green dashed box shows the nanograting.

Correspondingly, we modified the following:

1. We added the sentence "The blue dashed box shows dashed box shows the nanograting." in the updated manuscript on page 5, lines 83-84.
2. We revised a figure (Fig. 1f) in the updated manuscript on page 4, line 75.

[Comment 4-9]

8. *How to evaluate recognition accuracy, the definition of target current (signal) and training current (signal), etc., please give in SI.*

[Author reply 4-9]

We thank the Referee for the insightful comment and valuable suggestion. In our manuscript, the training current is defined as $I'_m = \sum_{n=1}^N I'_{mn} = \sum_{n=1}^N R'_{mn} P_n$, which is the output current of the array after training in a certain epoch. R'_{mn} is the photoresponsivity after being trained in a certain epoch, and P_n is the incident light power of the n th pixel with noise. The output current I_m that makes the activation function $\phi_m(I) = e^{I_m \xi} / \sum_{k=1}^M e^{I_k \xi}$ of the corresponding output node equal to 1 is defined as the target current. The recognition accuracy is defined as the maximum probability of the output node processed by the activation function after each epoch training, i.e. $\phi(I'_m) \times 100\%$, where I'_m is the training current.

Correspondingly, we modified the following:

1. We added the sentence "Here, target value and the training value." to the legend in Supplementary Fig. 9 in the updated Supplementary Information on pages 10-11.

[Comment 4-10]

9. The author claims that the device is a hardware solution for artificial vision with both sensing and recognition functions. Please give specific information and corresponding meanings of each step of device training and image recognition in SI to help readers better understanding of the work. The author needs to show readers in detail the specific information of using the constructed device to complete image training and image recognition.

[Author reply 4-10]

We would like to thank the Referee for raising critical questions and valuable suggestions. In our proposed AVS, the pattern classification task is solved by a single-layer perceptron containing nine input neurons and one output neuron. The hardware implementation of a single-layer perceptron is accomplished by interconnecting 3×3 PPTs in an ANN manner to form a PPTA. The network is trained off-line using computer simulation, a method called the ex-situ training. Subsequently, the predetermined photoresponsivity matrix, that is, photoresponsivities scaled from dimensionless weights, is transferred to the PPTA to complete the image recognition. The corresponding weights can be written into the array by modulating the voltage V_{DS} , and the subsequently projected image generates corresponding output currents for each subpixel. The training processes of the ANN with experimental photoresponsivity curve are illustrated in Supplementary Fig. 14. The figure is also attached below. Figure 3g shows the total output

current of the array after each training epoch. The corresponding photoresponsivity of the array after each training epoch is also presented in Supplementary Fig. 15. The figure is also attached below. The currents clearly separate and stabilize after 100 epochs, with the largest current corresponding to the label of the projected letter.

Supplementary Figure 14: The training processes of the ANN with experimental photoresponsivity curve. The ANN is trained off-line according to the experimental photoresponsivity curve in Supplementary Fig. 10c. R_+ represents the positive potentiation part of the photoresponsivity curve, while R_- represents the negative depression part of the photoresponsivity curve.

Fig. 3g. The total output current of the projected letter after each training epoch, where the letter ‘z’ is projected onto the chip. The maximum current corresponds to the label of the projected letter.

Supplementary Figure 15: Photoresponsivity and weight distributions of the array. a-d, The corresponding photoresponsivity of the array after 0 (a), 10(b), 30(c), and 50(d) training epochs, respectively. **e-f,** Photoresponsivity (e) and weight (f) distributions before (initial) and after (final) training.

Correspondingly, we modified the following:

1. We added two new figure (Supplementary Fig. 14 and Supplementary Fig. 15) in the updated Supplementary Information on pages 15-16.
2. We revised a figure (Fig. 3) in the updated manuscript on page 10.
3. We added an “Image recognition task” in the Methods section in the updated manuscript on page 20, lines 384-394.

List of main changes to the manuscript which are not covered by the point-by-point replies to the Referees:

1. We have added several authors who have contributed to the revision of the article, namely Xin Guo, Xinyi Fan, Zichen Wang, Yan Tong, and Decheng Wang.
2. We have added a Discussion section and several subheadings.

REVIEWER COMMENTS

Reviewer #1 (Remarks to the Author):

The authors have properly addressed all review comments and I would like to recommend acceptance.

Reviewer #2 (Remarks to the Author):

The authors have fabricated an Ag nanograting and two-dimensional heterostructure integrated plasmonic phototransistor array, achieving a high-performance artificial visual system.

I have carefully reviewed authors' responses to my previous concerns and those raised by referee #4. I can recommend publication after the following revisions:

- 1.The authors should take more time to conduct systematic experiments on the individual device, provide detailed characterizations and give a Table to summarize the performance of the individual phototransistor. It is not hard to do this, but the authors should spend time analyzing them carefully.
- 2.The authors have conducted the ultrafast image recognition tests with their network array (Fig. 1f letters 'z' and 'u' in Supplementary Figure 19) and shown device arrays in Fig. 1f and Supplementary Figure 4. However, the clarity is not good. Readers may spend much time understanding how the authors use their network array for demonstration.
- 3.The word "system" in the title misleads readers. The title can be revised as "High performance artificial visual perception and recognition with plasmon-enhanced 2D material neural network" to avoid misleading.

Moreover, please address the following minor points to enhance the manuscript's overall quality.

- 1.Abstract Clarity: The abstract contains an excessive use of abbreviations, affecting readability. Please consider simplifying the abstract by minimizing the use of technical terms to ensure broader comprehension.
- 2.Graph and Chart Color Scheme: Consider unifying the color scheme of graphs and charts to enhance visual consistency. A more straightforward and unified color palette can contribute to better overall aesthetics.
- 3.Table 1 Modifications: Table 1 requires revisions, and it's worth considering whether it should be included in the main text. The question marks ("OSR ?" and "CC?") in the table header appear to be

typos. If included in the main text, simplify the table; references (Ref.) can be directly cited in the first column without the need for a separate column.

4. Conclusion Section Enhancement: The conclusion section is somewhat generic. Please consider incorporating specific performance metrics and data to provide a more comprehensive summary of research findings.

Reviewer #3 (Remarks to the Author):

I read the revised manuscript carefully. The authors have considered my earlier comments and modified the manuscript accordingly. I have no further comment. The paper can now be published.

Reviewer #4 (Remarks to the Author):

I have read the revised manuscript and the author's response. Now I can clearly give my conclusion. As many serious problems remain with this work, I do not consider this manuscript suitable for publication in Nature Communications.

1. The performance of individual phototransistors is very important for further research on networks. However, in this work, the authors failed to analyze and test the basic optoelectronic properties of the unit device.
2. The paper has not yet completed the construction of the hardware network based on the phototransistor array. Note that this is a real network array, not several individual transistors. From this point, the authors are not using hardware to demonstrate the capabilities of neural networks at a real array level.
3. The paper shows the visual function of the device, but there is still a big gap between the current device array and the visual system with only partial functions. Calling this work a high-performance vision system in the title is unreasonable.

Table of Contents

– Responses to Reviewer #2’s comments: page 1-11

=====

Responses to Reviewer #2’s comments (NCOMMS-23-32413A)

=====

[Comment 2-1]

The authors have fabricated an Ag nanograting and two-dimensional heterostructure integrated plasmonic phototransistor array, achieving a high-performance artificial visual system.

I have carefully reviewed authors' responses to my previous concerns and those raised by referee #4. I can recommend publication after the following revisions:

[Author reply 2-1]

We appreciate the Referee's valuable time to evaluate our manuscript and make an expert summary. We are very grateful to Referee #2 for his/her previous thoughtful comments and suggestions, which we think have helped us greatly to improve the readability and clarity of our manuscript. In response to the Referee’s concerns as well as various suggestions, we have updated and revised the paper accordingly, which are summarized below in a point-to-point fashion.

[Comment 2-2]

1. The authors should take more time to conduct systematic experiments on the individual device, provide detailed characterizations and give a Table to summarize the performance of the individual phototransistor. It is not hard to do this, but the authors should spend time analyzing them carefully.

[Author reply 2-2]

We appreciate the useful comment and suggestion from the Referee. In addition to the transfer characteristic curves of the phototransistor under a large voltage range presented in the maintext, the transfer characteristic curves of the phototransistor corresponding to different optical wavelengths at relatively small voltage ($V_{DS} = 0.1$ V) are also presented in Supplementary Figs. 23a-c. The figure is also attached below. Similar to the

pre-processing under large gate voltage modulation, the features of the image can also be clearly presented under small gate voltage modulation, as shown in Supplementary Figs. 23d-f, although the clarity is usually weaker than that under large gate voltage modulation.

Supplementary Figure 23: Transfer characteristic curve and its application in image pre-processing process. **a-c**, The transfer characteristic curves of the devices with red (a) green (b) and blue (c) light measured under different P values at $V_{DS} = 0.1$ V, respectively. **d-f**, The pre-processing process corresponds to the red letter ‘z’ (d), the green letter ‘j’ (e) and the blue letter ‘u’ (f), respectively. The left column represents the image with Gaussian noise ($\sigma = 0.4$) added before pre-processing, the middle column represents the modulation voltage required for pre-processing, and the right column represents the image after pre-processing. Here, the modulation voltage in parentheses is the drain-source voltage V_{DS} .

Supplementary Table 1 Summary of the performance of individual plasmonic phototransistor.

Type	Mobility: light on/off ($\text{cm}^2 \text{V}^{-1} \text{s}^{-1}$)	On/off ratio	t (ns)	R (pA/μW)	Leakage current (pA)	Dark current (pA)
Red	154 / 23.3	$\sim 1 \times 10^8$	500	-14/+14	~ 10.2	~ 1.5
Green	233 / 23.4	$\sim 1 \times 10^8$	500	-16/+16	~ 10.2	~ 3.1
Blue	273 / 23.6	$\sim 1 \times 10^9$	500	-47/+47	~ 10.2	~ 2.3

t: response time. **R**: Photoresponsivity.

Based on the measured data, we summarize the performance of individual phototransistors in Supplementary Table 1. The table is also attached below. An analysis on the performance of individual plasmonic phototransistor is provided in Supplementary Note 2. According to the data presented in Fig. 3h, the mobility of the device at maximum optical power incidence is $154 \text{ cm}^2 \text{ V}^{-1} \text{ s}^{-1}$, which can be extracted using the expression $\mu = [dI_D/dV_G] \times [L/(WC V_{DS})]$, where $L = 1.9 \text{ }\mu\text{m}$ is the channel length, $W = 4.2 \text{ }\mu\text{m}$ is the channel width, and C is the capacitance between the channel and the gate per unit area ($C = \epsilon_0 \epsilon_r / d_{\text{hBN}}$; $\epsilon_r = 3.5$; $d_{\text{hBN}} = 10 \text{ nm}$). Similarly, the mobility of the device under green and blue light incidence with maximum optical power is $233 \text{ cm}^2 \text{ V}^{-1} \text{ s}^{-1}$ and $273 \text{ cm}^2 \text{ V}^{-1} \text{ s}^{-1}$, respectively, which can also be derived from the data presented in Supplementary Fig. 16. It is clear that the mobility of 2D materials has been greatly improved. On the one hand, the increase in mobility could be due to the suppression of Coulomb scattering by the deposited dielectric Al_2O_3 . On the other hand, the dephasing of plasmon generated by the Ag nanograting via photoexcitation leads to an increase in electron temperature and lattice temperature. Since the relaxation time of hot electrons is proportional to the temperature increment ΔT , the relaxation time of hot electrons increases. According to $\mu_e = e\tau/m^*$, where e is the elementary charge, τ is the relaxation time, and m^* is the effective mass, the carrier mobility is greatly improved. As shown in Supplementary Fig. 10a, the normalized transmittance spectrum of the WPPs structure in the device indicates that the absorption rate of the device increases as the wavelength decreases. That is, at the same optical power, the nanostructure in the device absorb blue light the most, followed by green light and finally red light. Therefore, the mobility of the device under green light incidence is maximum. Conversely, in the absence of light, the temperature of the nanostructure in the device remains unchanged, resulting in a significant decrease in mobility.

For the device with nanograting, under illumination, the dephasing of plasmons not only increases the device temperature, but also leads to a significant increase in electron concentration n in the device. According to $\sigma = ne\mu$, it is evident that the conductivity σ of the device is enhanced, and the on-state current is subsequently increased, ultimately resulting in a current on/off ratio exceeding 1×10^9 . It is also due to the introduction of nanogratings that the photoresponsivity of the device has been greatly improved. Although the entire process from the generation of hot electrons by plasmon decay to the injection into MoS_2 is accomplished on a sub-nanosecond level, the transfer of the hot electrons and the establishment of the thermoelectric potential prolong the entire process. In order to illustrate the high-speed capabilities of PPTA, we carried out measurements by employing a 500 ns pulsed laser source and an electric pulse source with synchronous triggering. As shown in Supplementary Fig. 19, we plot the electric output pulses, with different output codes representing different image types, which demonstrate the correct pattern classification within $\sim 500 \text{ ns}$. The dielectric h-BN between the channel WSe_2 and the gate MoS_2 derived from bulk source materials by a mechanical peel-transfer method. The weak leakage current indicates that

BN has good uniformity, insulation, and clean interface. Meanwhile, the negligible dark current indicates the absence of impurities and interface charges at the heterostructure interface, which is also verified in the scanning transmission electron microscope image shown in Fig. 1g.

Correspondingly, we modified the following:

1. We added the sentence "Besides, the transfer Supplementary Note 2, respectively." in the updated manuscript on page 12, lines 237-241.
2. We added the sentence "Similarly, the features large gate voltage modulation." in the updated manuscript on pages 12-13, lines 247-250.
3. We added a "Supplementary Note 2" section in the updated Supplementary Information on page 27.
4. We added a new figure (Supplementary Fig. 23) in the updated Supplementary Information on page 23.
5. We added a new table (Supplementary Table 1) in the updated Supplementary Information on page 23.

[Comment 2-3]

2. The authors have conducted the ultrafast image recognition tests with their network array (Fig. 1f letters 'z' and 'u' in Supplementary Figure 19) and shown device arrays in Fig. 1f and Supplementary Figure 4. However, the clarity is not good. Readers may spend much time understanding how the authors use their network array for demonstration.

[Author reply 2-3]

We thank the Referee for this useful comment. To make the whole process of ultrafast image recognition based on the network array more explicit, new Supplementary Figs. 19a, b are introduced for demonstration. The figure is also attached below. As shown in Supplementary Fig. 19a, each pixel contained in the image is illuminated on the plasmonic phototransistor array with a pulsed laser at a different power P_N . Upon optical stimulation, a total output current I_M is generated by a circuit in the array consisting of all the M th subpixels connected in a neural network manner. Subsequently, the generated current I_M is amplified by the preamplifier and converted into voltage V_M input into the oscilloscope. The voltage signals corresponding to the different letters 'z' and 'u' are presented in Supplementary Figs. 19c, d. The principle of generating total current I_M is displayed in Supplementary Fig. 19b. During the sensing process, the current generated by an individual pixel being stimulated can be obtained by multiplying the optical power and

photoresponsivity. The total output current I_M is then achieved by accumulating the currents generated by all the subpixels belonging to the same group along the interconnected sensory elements through Kirchhoff's law. For an array containing $M \times N$ pixels, with vector \mathbf{P} representing the optical stimulus and R_{MN} representing the photoresponsivity matrix, then the output current vector \mathbf{I} can be expressed as:

$$\mathbf{I} = R\mathbf{P} = \begin{bmatrix} I_1 \\ I_2 \\ \vdots \\ I_M \end{bmatrix} = \begin{bmatrix} R_{11} & R_{12} & \dots & R_{1N} \\ R_{21} & R_{22} & \dots & R_{2N} \\ \vdots & \vdots & \ddots & \vdots \\ R_{M1} & R_{M2} & \dots & R_{MN} \end{bmatrix} \begin{bmatrix} P_1 \\ P_2 \\ \vdots \\ P_N \end{bmatrix}$$

Stimulated by ultrafast signals corresponding to different subpixel groups in the array, different output currents I_m are generated, where $m = 1, 2, \dots, M$. Finally, different output currents are amplified and converted into voltages, which are then read out by an oscilloscope.

Supplementary Figure 19: Ultrafast image recognition. **a**, Flowchart for ultrafast image recognition. P_N represents the ultrafast signal incident on each pixel, I_M is the total output current of all M th subpixels, and V_M is the voltage obtained by converting and amplifying I_M through a preamplifier. **b**, Schematic diagram of a PPTA used to perform multiply-and-accumulation (MAC) operations in a neural network. R_{MN} is the regularized photoresponsivity of the subpixel. **c, d**, Projection of two different letters, ‘z’ and ‘u’, with a duration of 500 ns, leads to distinct output voltage codes 110 (c) and 111 (d) for the three channels. CH represents the output channel. The encoding rule for output voltage are the same as in the maintext, with positive signals marked as 1 and negative signals marked as 0.

Correspondingly, we modified the following:

1. We added the sentence " As shown in Supplementary Fig. 19a is displayed in Supplementary Fig. 19b." in the updated manuscript on page 15, lines 287-291.
2. We revised a figure (Supplementary Fig. 19) in the updated Supplementary Information on page 19.

[Comment 2-4]

3. The word “system” in the title misleads readers. The title can be revised as “High performance artificial visual perception and recognition with plasmon-enhanced 2D material neural network” to avoid misleading.

[Author reply 2-4]

We appreciate the critical comment and suggestion from the Referee. We agree to change the title of the manuscript to "High performance artificial visual perception and recognition with plasmon-enhanced 2D material neural network".

Correspondingly, we modified the following:

1. We revised the title of the manuscript in the updated manuscript on page 1, line 1.

[Comment 2-5]

Moreover, please address the following minor points to enhance the manuscript's overall quality.

1. Abstract Clarity: The abstract contains an excessive use of abbreviations, affecting readability. Please consider simplifying the abstract by minimizing the use of technical terms to ensure broader comprehension.

[Author reply 2-5]

We appreciate the Referee's valuable time to comment on our manuscript and provide valuable suggestion. In response to the Referee's concern, we replaced some abbreviations with more comprehensible words to ensure the readability of the manuscript.

Correspondingly, we modified the following:

1. We replaced "Artificial visual systems (AVS)" with "Neuromorphic visual systems" in the updated manuscript on page 1, line 12.
2. We replaced "complementary metal oxide semiconductor (CMOS) platform" with "silicon technology" in the updated manuscript on page 1, line 14.
3. We replaced "AVS" with "neuromorphic vision chips" in the updated manuscript on page 1, line 17.
4. We replaced "AVS" with "architecture" in the updated manuscript on page 1, line 17.
5. We replaced "AVS" with "AVPRM" in the updated manuscript.

[Comment 2-6]

2. Graph and Chart Color Scheme: Consider unifying the color scheme of graphs and charts to enhance visual consistency. A more straightforward and unified color palette can contribute to better overall aesthetics.

[Author reply 2-6]

We thank the Referee for the useful suggestion. To ensure consistency in the color scheme, we adjusted the colorbar in Fig. 3e to match the colorbar in Fig. 3f. We also adjusted the green curve in Fig. 3c and Fig. 3h to gray curve to match the color of the curve in Fig. 3g. The figure is also attached below.

Fig. 3 Functional implementation of 2D PPTA. **a** Multi-state photocurrents corresponding to different levels of optical power (gray levels), where the laser wavelength is 635 nm and the drain-source voltage is 0.1 V. **b** Extraction of the gray level of each pixel in the image. The images were added with Gaussian noise (with standard deviation of $\sigma = 0.3$). **c** Photocurrent of different colors of light (R: 635nm, G: 532nm, B: 473nm) under the same measurement conditions, where the power is $10 \mu\text{W}$ and the drain-source voltage is 0.1 V. **d** Recognition of different color images. The numbers in different colored boxes represent the maximum gray level extracted from (b) and its corresponding current. **e** The voltage (V_{DS}) tunable photocurrent corresponding to each gray level. **f** The photoresponsivity of the array and its modulation process after 100 off-line training epochs of the letter 'z' in (d). **g** The total output current of the projected letter after each training epoch, where the letter 'z' in (d) is projected onto the chip. The maximum current corresponds to the label of the projected letter. **h** The transfer characteristic curves of the devices with red light measured under different P values at $V_{DS} = 1$ V, respectively. **i** The pre-processing process of images. The left column represents the image with Gaussian noise ($\sigma = 0.4$) added before pre-processing, the middle

column represents the modulation voltage required for pre-processing, and the right column represents the image after pre-processing.

Correspondingly, we modified the following:

1. We revised a figure (Fig. 3) in the updated manuscript on page 10.

[Comment 2-7]

3. *Table 1 Modifications: Table 1 requires revisions, and it's worth considering whether it should be included in the main text. The question marks ("OSR ?" and "CC?") in the table header appear to be typos. If included in the main text, simplify the table; references (Ref.) can be directly cited in the first column without the need for a separate column.*

[Author reply 2-7]

We appreciate the useful comment and suggestion from the Referee. In response to the Referee's suggestion, we moved Table 1 to "Supplementary Information" as Supplementary Table 2. To make the table more concise, we removed the question marks from the table header and cited the references (Ref.) in the first column. The table is also attached below.

Supplementary Table 2 Comparison of the proposed neuromorphic device with previous report.

Pixel structure	Integrated function	OSR	E	t	DR	CC
^[6] A hBN/WSe ₂ synaptic device + hBN/WSe ₂ photodetector	Sensing + postprocessing (pattern recognition)	N/A	66-532 fJ	10 ms	N/A	Yes
^[7] A WSe ₂ photodiode	Sensing + postprocessing (classify/encode images)	Yes	N/A	40 ns	N/A	Yes
^[8] A resistive pressure sensor +	Sensing (visual-haptic fusion) +	N/A	N/A	1 s	N/A	N/A

perovskite-based photodetector + hydrogel-based ionic cable and a synaptic transistor	postprocessing (pattern recognition)					
^[9] A MoS ₂ -pV3D3 phototransistor	Sensing + preprocessing	N/A	N/A	0.5 s	N/A	Yes
^[10] A synaptic device with a structure of Pd/MoO _x /ITO	Sensing + preprocessing (background noise reduction)	N/A	N/A	200 ms	N/A	Yes
^[11] A MoS ₂ phototransistor + UVO treatment	Sensing + preprocessing (Scotopic/Photopic adaptaton)	N/A	N/A	N/A	199 dB	Yes
A MoS ₂ /hBN/WSe ₂ plasmonic phototransistor	Sensing + preprocessing (image contrast enhancement) + postprocessing (image recognition)	Yes	2.4×10^{-17} J	500 ns	180 dB	Yes

DR stands for dynamic range and can be used to enhance image contrast (preprocessing) by adjusting DR. **E**: energy consumption. **t**: response time. **OSR**: On-site recognition. **CC**: CMOS compatibility.

Correspondingly, we modified the following:

1. We moved Table 1 to "Supplementary Information" as Supplementary Table 2 and modified it in the updated Supplementary Information on page 24.

[Comment 2-8]

4. *Conclusion Section Enhancement: The conclusion section is somewhat generic. Please consider incorporating specific performance metrics and data to provide a more comprehensive summary of research findings.*

[Author reply 2-8]

We appreciate the professional comment and valuable suggestion from the Referee. In response, we provided a more comprehensive summary based on the specific performance metrics and data of the device. We added the sentence "The strong coupling effect caused by the WPPs structure greatly enhances the absorption of the tricolor light in the device, thus improving the generation of hot electrons and the injection into the floating gate. Under the coordination of photothermoelectric effect caused by plasmon dephasing and electrical modulation, the current on/off ratio of the device exceeds $\sim 1 \times 10^9$ and the dynamic range reaches 180 dB. The performance of the device can greatly enhance the image contrast during the pre-processing process. Subsequent image recognition is successfully performed under the incidence of continuous light and pulsed light, respectively. Two letters with a duration of 500 ns can be recognized on the basis of consuming 2.4×10^{-17} J per spike." in the conclusion section.

Correspondingly, we modified the following:

1. We added the sentence "The strong coupling effect consuming 2.4×10^{-17} J per spike." to the Discussion section in the updated manuscript on pages 16-17, lines 329-336.

List of main changes to the manuscript which are not covered by the point-by-point replies to the Referees:

1. We added the sentence "In addition, the first prototype of artificial optical graded neuron was proposed and realized for processing spatiotemporal information with more than 99% accuracy" in the Introduction section and cited a new reference [43] in the References section.
2. We cited new references [6]-[11] in the Supplementary Information.

REVIEWERS' COMMENTS

Reviewer #2 (Remarks to the Author):

The authors have addressed my previous comments and I would like to recommend publication in Nature Communications.